# WHEN LLMS GET SIGNIFICANTLY WORSE: A STATISTICAL APPROACH TO DETECT MODEL DEGRADATIONS

**Jonas Kübler, Kailash Budhathoki, Matthäus Kleindessner, Xiong Zhou, Junming Yin,
Ashish Khetan, George Karypis**
Amazon
{kuebj,kaibud,matkle,xiongzho,junmingy,khetan,gkarypis}
@amazon.com

## ABSTRACT

Minimizing the inference cost and latency of foundation models has become a crucial area of research. Optimization approaches include theoretically lossless methods and others without accuracy guarantees like quantization. In all of these cases it is crucial to ensure that the model quality has not degraded. However, even at temperature zero, model generations are not necessarily robust even to theoretically lossless model optimizations due to numerical errors. We thus require statistical tools to decide whether a finite-sample accuracy deviation is an evidence of a model's degradation or whether it can be attributed to (harmless) noise in the evaluation. We propose a statistically sound hypothesis testing framework based on McNemar's test allowing to efficiently detect model degradations, while guaranteeing a controlled rate of false positives. The crucial insight is that we have to confront the model scores on each sample, rather than aggregated on the task level. Furthermore, we propose three approaches to aggregate accuracy estimates across multiple benchmarks into a single decision. We provide an implementation on top of the largely adopted open-source LM Evaluation Harness and provide a case study illustrating that the method correctly flags degraded models, while not flagging model optimizations that are provably lossless. We find that with our tests even empirical accuracy degradations of *0.3%* can be confidently attributed to *actual* degradations rather than noise. https://github.com/amazon-science/LLM-Accuracy-Stats

## 1 INTRODUCTION

The widespread adoption of Large Language Models (LLMs) has led to a surge in new approaches to accelerate their inference and reduce their usage cost. The spectrum of such optimizations is wide (Zhou et al., 2024; Park et al., 2024), ranging from optimizations that leave the computed function untouched like efficient software stacks (vLLM Contributors, 2024) and efficient kernels (Dao et al., 2022) over optimizations that change the function but provably maintain the output distribution like speculative decoding (Chen et al., 2023), all the way to optimizations that change (potentially very slightly) the model's output distribution like quantization (Kurtic et al., 2025; Frantar et al., 2023) or sparsity (LeCun et al., 1989; Hubara et al., 2021; Frantar & Alistarh, 2023; Sun et al., 2024).

On paper, the distinction between *theoretically lossless* methods and *lossy* methods seems clear, however, modern LLMs are usually served at maximum in 16-bit precision for activations, some are already trained with 8-bit matrix weights (Liu et al., 2024) and the recent *gpt-oss* is even natively quantized in a mixed-precision 4-bit datatype (Agarwal et al., 2025). In this regime the non-associative nature of floating-point arithmetic, where, e.g., $(a + b) + c \neq a + (b + c)$, accumulates errors throughout the computational pipeline. In practice, this manifests in the observation that the same LLM can generate different responses depending on the hardware, the framework, and even the batch in which a request is run. This happens even when the sampling temperature is set to zero, where, again on paper, one would expect them to be fully deterministic, see Yuan et al. (2025) and a

recent blog of thinkingmachines.ai.[1] This suddenly blurs the line between *theoretically lossless* and *lossy* optimizations.

The purpose of this paper is to develop a rigorous statistical framework to quantify the uncertainty of such deviations and to decide whether an observed deviation is based on statistical fluctuations or whether the model got actually worse. We rigorously define the problem of detecting accuracy degradation against a baseline model (and inference stack) and describe the common pitfalls in quantifying their uncertainty estimates. We identify that McNemar (1947) introduced a test for a similar situation; we adapt and refine this test to provide higher test power for our setting. Our theoretical analysis of the asymptotic test power provides a recommendation on how to compress existing datasets for efficient degradation detection. We then introduce three variants to aggregate tests when evaluating on multiple datasets and illustrate their pros and cons through synthetic experiments. We provide a script to run the proposed statistical tests on top of the widely adopted LM Evaluation Harness and provide extensive experiments on LLMs showing that our testing framework succeeds in detecting even empirical accuracy degradations of 0.3% as significant. To facilitate the reading, we provide an overview of our notation and the introduced algorithms in Appendix A.

## 2 BACKGROUND AND RELATED WORK

### 2.1 ACCURACY ESTIMATION

Let $\mathcal{X}$ and $\mathcal{Y}$ be some input/output domain, in our case texts of varying length. In benchmarking settings of LLMs commonly an input $x \in \mathcal{X}$ is fed to an LLM $M : \mathcal{X} \to \mathcal{Y}$ which produces an output $y \in \mathcal{Y}$.[2] The benchmarking task defines a scoring criterion $L : \mathcal{Y} \to \{0, 1\}$ that classifies the generation of the model as correct or incorrect, where for simplicity we focus on binary accuracy criteria and consider 1 as success and 0 as failure to solve the problem $x$. We present a generalization to non-binary scores in Appendix D. Together with a probability distribution $P_X$ over the input domain $\mathcal{X}$, we can define the *accuracy* of model $M$ as

$$\gamma := \mathbb{E}_{X \sim P_X} L(M(X)). \tag{1}$$

Now in practice, we do not have full access to the distribution $P_X$. Instead we have a set of $N \in \mathbb{N}$ examples that are potentially hand curated. For the statistical treatment, we model this situation by assuming that the examples are independently and identically distributed and all drawn from $P_X$, i.e., $x_1 \ldots, x_N \overset{\text{i.i.d.}}{\sim} P_X$. From the finite sample we estimate the empirical accuracy as $\hat{\gamma} = \sum_{i=1}^N L\left(M(x_i)\right)/N$. Conceptually, we are estimating the mean of a *Bernoulli* random variable, the estimate $\hat{\gamma}$ has variance $\text{Var}[\hat{\gamma}] = \frac{\gamma(1-\gamma)}{N}$, and the number of successes ($N\hat{\gamma}$) follows a binomial distribution. In this work we are assessing whether an optimized model $\tilde{M}$, for which we define the population accuracy as $\beta$ and its finite sample estimate as $\hat{\beta}$, has degraded accuracy. We thus want to test the *null hypothesis* $H_0 : \beta = \gamma$ against the *alternative hypothesis* $H_A : \beta < \gamma$. On first sight this might seem as trivial as deciding whether two coins have the same probability of heads or not. However, in LLM benchmarking there is one crucial difficulty: we are using the same examples $x_1 \ldots, x_N$ to estimate $\hat{\gamma}$ and $\hat{\beta}$. Thus our accuracy estimates are *not* statistically independent, the same problem as in before-after tests on the same set of patients in medical treatment testing. If the accuracy estimates were independent, the variance of the difference $\hat{\gamma} - \hat{\beta}$ would simply be the sum of the two variances. However, due to the dependence it is not just a function of the separate variances alone and in particular

$$\text{Var}[\hat{\gamma} - \hat{\beta}] \neq \frac{\gamma(1-\gamma)}{N} + \frac{\beta(1-\beta)}{N}. \tag{2}$$

For our further analysis we define the bivariate random variable

$$L_{M,\tilde{M}}(X) := \left(L(M(X)), \, L(\tilde{M}(X))\right), \quad \text{where } X \sim P_X, \tag{3}$$

i.e., the distribution of $L_{M,\tilde{M}}$ is a push-forward distribution of $P_X$. It has values in $\{0, 1\}^2$ and we define its probabilities as $P_a = P(0,0)$, $P_b = P(1,0)$, $P_c = P(0,1)$, $P_d = P(1,1)$.

---

[1] https://thinkingmachines.ai/blog/defeating-nondeterminism-in-llm-inference/

[2] The output generation of the LLM can be probabilistic itself, for example when the generation temperature is non-zero or due to rounding errors. However, we could absorb any additional randomness formally by redefining $X$. Therefore, to keep the notation simple, we treat $M$ as a deterministic function.

## 2.2 Statistics and Hypothesis Testing

Our goal is to test *null hypothesis* $H_0 : \beta = \gamma$ against the *alternative hypothesis* $H_A : \beta < \gamma$. To do this we define a *test statistic*, which we compute on our observed data. We will derive the distribution of this test statistic under the null hypothesis and reject the null hypothesis if the observed test statistic is *unusually* large. We want to control two types of errors (Lehmann & Romano, 2005). If the null hypothesis is true, we want to ensure that the probability of false rejection is controlled at $\alpha \in (0, 1)$. Although there is no deeper meaning to $\alpha = 5\%$ this is the default choice in many papers and we will follow that. If we falsely reject $H_0$ we call this a type-I error. Correctly characterizing the null distribution of the test statistic and rejecting observations if and only if they are above the $1 - \alpha$ percentile, guarantees that the type-I error rate is controlled at $\alpha$ (Lehmann & Romano, 2005). We consider correct type-I error control as a necessary condition for a useful test. On the other hand, if the null hypothesis is false, we want our test to reject the null hypothesis as often as possible, to minimize the rate of type-II errors. In order to define reliable tests, it is crucial to correctly characterize the null distribution and in particular its variance.

In 1947 the statistician McNemar (McNemar, 1947) realized the same problem as in Equation (2): *"There are many situations in which the sampling variance of the difference between two proportions (or percentages) must take into account the fact that the two proportions are not based on independent samples."* Instead of treating the accuracy estimates as independent, we must assess how the two models agree or differ on the individual examples

Table 1: 2×2 Contingency Table

|  |  | $M$ | |
|---|---|---|---|
|  |  | 0 | 1 |
| $\tilde{M}$ | 0 | $a$ | $b$ |
|  | 1 | $c$ | $d$ |

$x_i$ in our benchmark. To facilitate this we collect the outcomes of the $N$ examples of the random variable $L_{M,\tilde{M}}$ in a contingency table, see Table 1, where $a$ counts the examples where both models failed, $b$ the examples where the baseline model $M$ succeeded and the optimized model $\tilde{M}$ failed etc., such that $N = a + b + c + d$. Furthermore, the accuracy estimates of the models are $\hat{\gamma} = (b + d)/N$ and $\hat{\beta} = (c + d)/N$. While the actual values of the accuracy estimates depend on $d$ and $a$, McNemar (1947) noted that to decide whether two proportions (in our case accuracies $\beta$ and $\gamma$) are the same, only the examples where the models disagree are relevant, i.e., $b$ and $c$. McNemar (1947) introduced the test statistic $\frac{(b-c)^2}{b+c}$ and derived that under the null hypothesis (and for fixed $b + c$) this follows approximately a chi-square distribution with one degree of freedom. This provided a reliable test to assess equivalence of proportions in non-independent examples.

## 2.3 LLM Accuracy Evaluation and Optimization

In the field of machine learning, McNemar's original test was previously used to assess which of either two models performs better on a single task (Meshbane & Morris, 1996; Rainio et al., 2024). Yang et al. (2025b) generalized McNemar's test to be used in repeated cross-validation settings to decide which algorithms performs better. However, their test derives loose bounds on the null hypothesis and is thus overly conservative. Heineman et al. (2025) discussed dataset properties that best allow to reliably tell models apart.

For inference optimization of LLMs we are not aware of work that used McNemar's test to detect accuracy degradation. For a general introduction on LLM evaluation, we refer to Chang et al. (2024). Kurtic et al. (2025) provide a comprehensive analysis of the accuracy and performance impact of quantizing LLMs with one of their main finding that *"FP8 (W8A8-FP) is effectively lossless across all model scales."* They do extensive accuracy evaluations and simply aggregate the accuracy numbers into a mean and state the difference of the means. Most publications investigating model compression techniques follow the same practice and simply provide the accuracy estimates (Frantar et al., 2023; Frantar & Alistarh, 2023; Sun et al., 2024; Kübler et al., 2025). While this provides some level of relative guidance it does not quantify the certainty of these estimates, which is crucial because of aforementioned correlation and even more obviously as the used benchmark datasets have very different sample sizes.

Libraries such as the *Language Model Evaluation Harness* (lm-eval) (Gao et al., 2024) actually provide estimates of the standard error for each task individually and they also provide a per-sample score. However, using those standard errors as independent when comparing two models, overestimates the error of the accuracy difference (Miller, 2024), see Figure 1 and Equation (2). The

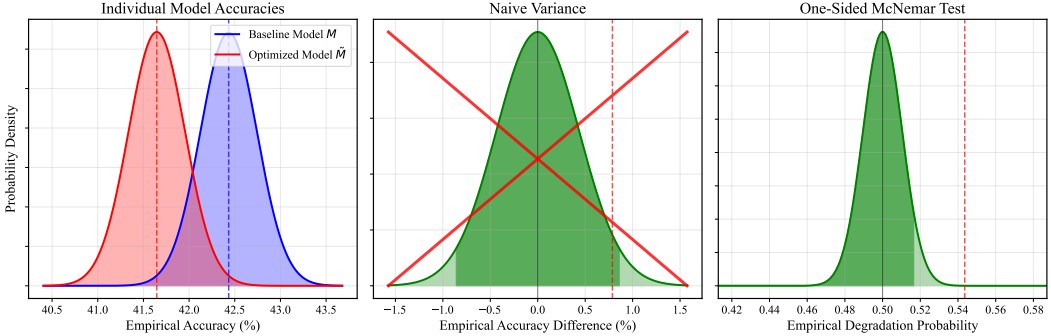

Figure 1: Detecting accuracy degradation on Llama-3.1 8B Instruct based on empirical estimates and their uncertainties. The optimized model $\tilde{M}$ quantizes the KV cache and the attention and results in an empirical accuracy drop of $0.79\%$ over different datasets (Section 4). When naively treating the accuracy estimates as independent, the sampling error of the difference is overestimated (Miller, 2024), hindering the detection of actual accuracy differences (middle). Instead, we advocate to compare the per-sample scores. Our proposed exact one-sided McNemar's test focuses on the *degradation probability*. In cases where there is no accuracy difference, the degradation probability is $0.5$ and we can exactly characterize its empirical distribution. This allows to detect small accuracy degradations as statistically significant, here with a $p$-value of 1.69e-05 (right).

*Open LLM Leaderboard* (Fourrier et al., 2024) provides a comprehensive set of accuracy benchmarks, that can all be run within the lm-eval package. Having accuracy estimates from multiple benchmarks provides the additional difficulty of deciding how to aggregate those numbers.

In a recent NeurIPS paper, Dutta et al. (2024) criticize that "*all [considered] quantization schemes have negligible difference in accuracy (0 – 2%) compared to the 16-bit version,*" yet the models changed in behavior. Dutta et al. (2024) thus conclude that accuracies are not a suitable means to detect model degradation. However, the $0 - 2\%$ is an arbitrary choice of theirs, without taking estimation errors into account. We will see that when the statistics are properly taken into account such deviations are often enough to conclude a *significant* degradation. Instead of accuracy, Dutta et al. (2024) propose to investigate the probability of a flip in scoring of the answer. While they show that this correlates well with model degradation, they miss the point that such flips also often occur even without optimizing the model, see Section 4. Furthermore, their approach lacks a quantification of the statistical relevance. Yuan et al. (2025) illustrate that for reasoning models the accuracy deviations in lossless cases are even larger than for non-reasoning models, reaching up to 9% accuracy deviation. While their focus is on characterizing the severity of such deviations and their root causes, their work further motivates the need for a proper statistical analysis of such deviations.

## 3 THEORY

### 3.1 EXACT ONE-SIDED MCNEMAR'S TEST

The random variable $L_{M,\tilde{M}}$ defined in Equation (3) links the two model evaluations of $M$ and $\tilde{M}$. Instead of thinking of two separate estimations, we can think of it as a joint accuracy experiment. We can define two probabilities:

$$p_{\updownarrow} := P_X[L(M(X)) \neq L(\tilde{M}(X))] = P_b + P_c \qquad \text{(flip),} \quad (4)$$

$$q_{\downarrow} := P_X[L(M(X)) = 1 \mid L(M(X)) \neq L(\tilde{M}(X))] = P_b/p_{\updownarrow} \quad \text{(conditional degradation).} \quad (5)$$

The flip probability quantifies how often the two models' scores disagree, i.e., that the scoring *flipped* (Dutta et al., 2024). Given that the two model predictions disagree, the (conditional) degradation probability quantifies how often the baseline model $M$ scored 1 and the optimized model $\tilde{M}$ scored 0. For conciseness we will drop "conditional". Furthermore, we have the following fact that we will use to formulate a crisp null hypothesis.

**Fact 1** (Model Degradation). *The accuracy $\beta$ of an optimized model is worse than the accuracy $\gamma$ of the original model if and only if the degradation probability $q_\downarrow$ is larger than $1/2$.*

The test statistic we propose is the empirical estimate of the degradation probability:

$$\hat{q_\downarrow} := \frac{b}{b+c}. \tag{6}$$

Since $b$ is the count of degradations out of $b + c$ flips, the distribution of $\hat{q_\downarrow}$ is directly linked to a binomial distribution, i.e., $\hat{q_\downarrow}(b+c) = b \sim \mathrm{Binomial}(b+c, q_\downarrow)$. By Fact 1, under the null hypothesis $q_\downarrow = 1/2$. To be more sensitive, we compute one-sided $p$-values, which is justified since we only test for *degradations* of the optimized model. For this we can use standard libraries like `scipy` (Virtanen & SciPy 1.0 Contributors, 2020) to compute exact $p$-values of an observation $\hat{q_\downarrow}$. In effect, our test is just a standard binomial test (Algorithm 1). To illustrate the relationship of our test to McNemar's original test, we note that $b + c$ is treated as a constant both in ours and McNemar's original test and obtain

$$\frac{(b-c)^2}{b+c} = \left( \frac{2b}{b+c} - \frac{b+c}{b+c} \right)^2 (b+c) = (2\hat{q_\downarrow} - 1)^2 (b+c). \tag{7}$$

Thus McNemar's original test statistic is purely the scaled square of an affine transformation of the degradation probability. Since our test is one-sided and allows for the exact computation of $p$-values, we call it *exact one-sided McNemar test*.[3]

## 3.2 NORMAL APPROXIMATION AND TEST POWER

Our test treats $b+c$ as a constant and only relies on the exact binomial distribution. However, to have a data- and cost-efficient test it also matters that out of $N$ overall examples not unnecessarily many are discarded. To build this understanding, in this subsection we analyze the accuracy degradation $\delta := \gamma - \beta = p_\updownarrow(2q_\downarrow - 1)$, which is directly linked to the degradation probability but also takes the flip probability into account. We will derive our insights based on an asymptotic analysis, which is a standard practice (Gretton et al., 2012),(Casella & Berger, 2002, Section 10.3.2).

For the accuracy difference $\delta$ we can define the function

$$D(x) = \begin{cases} 0 & \text{if } L(M(x)) = L(\tilde{M}(x)), \\ 1 & \text{if } L(M(x)) = 1 \text{ and } L(\tilde{M}(x)) = 0, \\ -1 & \text{if } L(M(x)) = 0 \text{ and } L(\tilde{M}(x)) = 1, \end{cases} \tag{8}$$

Such that $\mathbb{E}_{X \sim P_X}[D(X)] = \delta = P_b - P_c = 2p_\updownarrow(q_\downarrow - 1/2)$. Thus $\hat{\delta}$ is the sample mean of $D(x)$

$$\hat{\delta} = \sum_{i=1}^{N} D(x_i) = \frac{b-c}{N}. \tag{9}$$

The variance of the random variable $D$ is $\mathrm{Var}[D(X)] = P_b + P_c - (P_b - P_c)^2$, see Appendix B. Under the null hypothesis for which $q = 1/2$, we have $P_b = P_c$ and, quite curiously, the variance simplifies to the flip probability $\mathrm{Var}[D(X)] = P_b + P_c = p_\updownarrow$. For the accuracy difference based on $N$ samples we have $\mathrm{Var}[\hat{\delta}] = p_\updownarrow/N$.[4] Assuming $p_\updownarrow > 0$ and since $\hat{\delta}$ is an empirical mean, we can use a standard central limit theorem and Slutsky's theorem (Serfling, 1980) to derive the asymptotic distribution under the null hypothesis ($\delta = 0$, $\mathrm{Var}[\hat{\delta}] = \frac{p_\updownarrow}{N}$):

$$\sqrt{N/\hat{p_\updownarrow}}\, \hat{\delta} \xrightarrow[N \to \infty]{d} \mathcal{N}(0,1). \tag{10}$$

---

[3]We are not aware of any published scientific work that defines a one-sided McNemar test explicitly, but we do not claim it is novel, see also: https://stats.stackexchange.com/questions/341812/one-sided-mcnemars-test.

[4]If we were to ignore that the models are evaluated on the same data and hence correlated, the variance of $\hat{\delta}$ would be the sum of the variances of the individual accuracy estimates $\frac{2\gamma(1-\gamma)}{N}$, where we used that under the null hypothesis $\gamma = \beta$ (see also discussion around Equation (2)). For optimized LLMs, $\gamma$ is usually around 50% and $p_\updownarrow \approx 10\%$, see Section 4 and Dutta et al. (2024). For these example numbers we would have $\mathrm{Var}[\hat{\delta}] \approx \frac{1}{5}(\mathrm{Var}\hat{\gamma} + \mathrm{Var}\hat{\beta})$. This implies that in relevant scenarios for LLMs a naive approach that ignores the correlation would overestimate the uncertainty of the accuracy degradation by around $\sqrt{5} \approx 2.2$ and thus wrongly attribute actual degradations to statistical fluctuations, see Figure 1.

Thus, we could use $\sqrt{N/\hat{p_\updownarrow}}\,\hat{\delta}$ as test statistic and an asymptotic one-sided threshold would be $t_\alpha = \Phi^{-1}(1 - \alpha)$, where $\Phi$ denotes the CDF of a standard normal, $\Phi^{-1}$ its inverse. This is exactly the common $z$-test (Casella & Berger, 2002, Section 10.3.2). Now given a certain model degradation $q_\downarrow > 1/2$ the test power can also be derived in closed form and is (Gretton et al., 2012, Section 3)

$$1 - \Phi\left(t_\alpha - \sqrt{N/p_\updownarrow}\,\delta\right). \tag{11}$$

So the test power is solely determined by the signal to noise ratio SNR $:= \sqrt{N/p_\updownarrow}\,\delta$. Now let us assume that we modify our dataset to contain only the examples where the models disagree (we can emulate this by setting $a = 0$ and $d = 0$ post-hoc). Then we have $\delta \to \delta' = \frac{\delta}{p_\updownarrow}$, $N \to N' = Np_\updownarrow$, $p_\updownarrow \to p_\updownarrow' = 1$, i.e., the accuracy difference increases proportionally, the effective sample size gets reduced and the new flip probability is 1. For the updated SNR, however, we have

$$\text{SNR}' := \sqrt{N'}\frac{\delta'}{\sqrt{p_\updownarrow'}} = \sqrt{Np_\updownarrow}\frac{\delta/p_\updownarrow}{1} = \sqrt{N}\frac{\delta}{\sqrt{p_\updownarrow}} = \text{SNR}, \tag{12}$$

i.e., the SNR remains unchanged even if we remove examples where both models agree. Since evaluations are costly and take time, this theoretical insight leads to the following:

**Recommendation 1.** *Examples that are unlikely to flip should be removed from datasets for degradation analysis.*

If one is to interpret actual accuracy numbers, those examples matter too. But when assessing whether a model has significantly degraded or not, they add cost, but no signal. Dutta et al. (2024, Section 5) have analyzed which examples are likely to flip in classification tasks based on the margin of the highest probability token. In Section 4.1 we elaborate more on this and generalize this to generative tasks. We evaluate an approach that allows us to reduce the sample size of MMLU Pro by almost 50% while keeping most of the flipping examples that are relevant to detect degradation.

Our previous discussion leads to the recommendation to redefine dataset in order to increase their flip probability (and thus also scale the empirical accuracy degradation). This underlines once more that a degradation analysis based on fixed accuracy thresholds or flip probabilities cannot be robust across models and datasets.

As a user, one might care more about the actual accuracy drop $\delta$ rather than the degradation probability $q_\downarrow$, so one would ideally like to answer questions like "How large do I need to choose $N$ such that I can detect a $0.2\%$ accuracy drop with probability $95\%$?" However, the test power Equation (11) is not only a function of $N$ and $\delta$ but also of the flip probability $p_\updownarrow$. Thus, we cannot answer such questions in general. In fact, for a fixed accuracy degradation $\delta$ the probability of detecting it decreases as $p_\updownarrow$ grows. While this might seem counterintuitive at first sight, using $\delta = 2p(q_\downarrow - 1/2)$ we get SNR $= \sqrt{N}\frac{\delta}{\sqrt{p_\updownarrow}} = 2\sqrt{Np_\updownarrow}(q_\downarrow - 1/2)$, which grows as the degradation probability increases. We illustrate this in Figure 4.

### 3.3 Aggregation across Tasks

When evaluating models across multiple tasks $T \in \mathbb{N}$, the question arises how to aggregate the observations into a single $p$-value. Effectively, we want to test the null hypothesis $H_0 : \gamma_i = \beta_i \,\forall i \in [T]$ against the alternative hypothesis $H_A : \exists i \in [T] : \gamma_i > \beta_i$. As in Section 2.2 we collect contingency tables for each task $i$, yielding counts $b_1, \ldots, b_T$ and $c_1, \ldots, c_T$. We propose three statistical tests that aggregate these individual task results (see Appendix A):

**Pooled Test:** We define $b = \sum_i b_i$ and $c = \sum_i c_i$, then apply the exact one-sided McNemar test as if all tasks formed a single large dataset. This approach naturally weights tasks by their sample sizes, providing high sensitivity when degradation affects multiple tasks homogeneously. On the other hand having tasks with large sample counts that do not provide signal, i.e., for which $b_i \approx c_i$, will harm this test. One big advantage of the pooled test is that it is robust to subgrouping of tasks, because every sample will always have the same weight. The following two tests instead will lead to varying $p$-values depending on how subtasks are defined and whether they are grouped together.

**Max Drop Test:** The purpose of this test is to identify whether the most significant observed degradation is significant overall. This is particularly useful if we suspect that an optimization might only

affect a single of our benchmarks. For each task $i$, we therefore calculate the standardized degradation statistic $\hat{z}_i = \frac{\hat{q}_{\downarrow i} - 0.5}{\text{SE}(\hat{q}_{\downarrow i})}$, where $\text{SE}(\hat{q}_{\downarrow i}) = \sqrt{0.25/(b_i + c_i)}$ is the standard error for a given task and takes into account that the task might have different overall counts or numbers of flips. Following Section 3.2 this is an estimate of the SNR for each task. By our test power consideration, we then select the task, which seems most promising for detecting an accuracy drop. Our test statistic is thus $z_{\max} = \max_i \hat{z}_i$. Since the null distribution is complex, we use Monte Carlo simulation: for each round, we generate $b_i^{\text{sim}} \sim \text{Binomial}(b_i + c_i, 0.5)$, compute the corresponding $z_i^{\text{sim}}$ values, and calculate $z_{\max}^{\text{sim}}$. The $p$-value is the proportion of simulated statistics $\leq$ our observed value.

**Fisher's Method:** Since we assume that all tasks are based on independent samples, we can apply the standard Fisher's method (Fisher, 1932). We first compute $p$-values $p_i$ for each task using the exact one-sided McNemar test. Fisher's method combines these $p$-values using the test statistic $\chi^2 = -2 \sum_{i=1}^{T} \ln(p_i)$. Under the null hypothesis, this statistic follows a chi-squared distribution with $2T$ degrees of freedom, allowing exact $p$-value computation. This approach provides a balanced sensitivity between the pooled test and max drop test.

**Combined Decision:** We have proposed three variants to obtain $p$-values to test the null hypothesis of accuracy degradation. Since those $p$-values are statistically dependent (as opposed to the individual task $p$-values we aggregated via Fisher's method), we cannot tightly combine them into a single $p$-value. Instead we propose to consider the simple decision rule to flag accuracy degradation when any of the three tests rejects at $\alpha$. Formally, via the argument of Bonferroni correction, this controls type-I error at $3\alpha$, however, as we will see from our experiments in Figure 3 the effective type-I error control usually happens even tighter than $3\alpha$.

In Appendix 3 we include synthetic experiments generating scenarios for null hypothesis ($q_\downarrow = 1/2$ for all tasks for varying $T \in \mathcal{N}$) and alternative hypothesis ($q_\downarrow > 1/2$ for all or a single out of varying number of $T \in \mathcal{N}$ tasks) and demonstrate our theoretical insights and the pros and cons of the different aggregation strategies. In particular, we find that the pooled test is most powerful in scenarios where the degradation happens across tasks, whereas the max drop test is best when only a single task is affected. Fisher's method strikes a good balance between these two extremes.

## 4 LLM EXPERIMENTS

We apply our proposed tests on different configurations of Llama-3.1 8B Instruct and Llama-3.3 70B Instruct (Dubey et al., 2024) as well as Mistral[5] Small 3.1. Furthermore, we also evaluate a pruned and distilled variant of the non-instruct Llama-3.1 8B model. We run the Leaderboard v2 (Fourrier et al., 2024) benchmark suite, which contains Big Bench Hard (BBH) (Suzgun et al., 2022), GPQA (Rein et al., 2023), IFEval (Zhou et al., 2023), MATH hard (Lewkowycz et al., 2022; Hendrycks et al., 2021), MMLU-pro (Wang et al., 2024), and Musr (Sprague et al., 2024). The sample sizes are quite different, see top of Table 5, which necessitates proper statistical treatment. The total number of examples in this suite is 25,282. For the 70B model we ignore the MATH task, as even the baseline scores close to zero due to some parsing mismatch, reducing the overall sample size to 20,282 for the 70B model.

We use vLLM (vLLM Contributors, 2024) v0.10.0 and lm-eval v0.4.8. Our baseline is the official BF16 model served with tensor parallelism 8 on 8 H100 GPUs. The results of our statistical analysis are shown in Table 2 and in the appendix we include the full accuracy numbers (Table 5) and contingency tables (Table 6, Table 7, Table 8, Table 9). While we do not want to recommend 5% as a general significance level used in practice, we use it here to classify "statistically significant" degradations when either of the three $p$-values is below this threshold.

**Baselines:** We are not aware of any explicitly stated rules to detect statistically significant accuracy deviation. We use the two loosely defined rules by Dutta et al. (2024, Section 1) i) "difference in aggregate accuracy metric [...] is negligible [...] ($\leq 2\%$)" that is $\hat{\delta} > 2\%$ means degradation and ii) "the actual percentage change [flip probability] in the answer can be significant ($\geq 5\%$)" that is $\hat{p}_\updownarrow \geq 5\%$ indicates degradation.

---

[5]https://mistral.ai/news/mistral-small-3-1

Table 2: Statistical analysis of accuracy degradations on Llama-3.1 8B Instruct, Llama-3.3 70B Instruct, Mistral-Small-3.1-24B-Instruct-2503, and Llama-3.1 8B (non-instruct). We report the pooled accuracy estimate over all tasks from Table 5 $\hat{\gamma}$, the accuracy difference against the baseline $\hat{\delta}$ and its standard error $\text{SE}(\hat{\delta})$. Further, the three $p$-values we propose and the probability of flips.

| Model | $\hat{\gamma}$ | $\hat{\delta}$ | $\text{SE}\hat{\delta}$ | $p_{\text{pool}}$ | $p_{\text{max drop}}$ | $p_{\text{fisher}}$ | $\hat{p}_{\updownarrow}$ |
|---|---|---|---|---|---|---|---|
| Baseline 3.1 8B | 42.43% | – | – | – | – | – | – |
| Baseline (rerun) | 42.45% | -0.02% | 0.07% | 6.29e-01 | 8.04e-01 | 8.68e-01 | 1.31% |
| transformers | 42.51% | 0.079% | 0.10% | 7.87e-01 | 7.144-01 | 8.66e-01 | 2.76% |
| TP1 | 42.45% | -0.01% | 0.10% | 5.64e-01 | 7.58e-01 | 8.88e-01 | 2.40% |
| A100 | 42.58% | -0.14% | 0.10% | 9.21e-01 | 9.93e-01 | 9.81e-01 | 2.73% |
| w4a16 | 40.70% | 1.73% | 0.22% | **4.80e-15** | **0.00e+00** | **2.84e-15** | 12.63% |
| FP8 | 42.48% | -0.04% | 0.19% | 6.01e-01 | 2.82e-01 | 5.49e-01 | 8.71% |
| w8a16 | 42.32% | 0.11% | 0.13% | 2.11e-01 | **1.36e-02** | **1.33e-02** | 4.45% |
| FP8-dynamic | 42.39% | 0.05% | 0.17% | 4.01e-01 | 4.29e-01 | 5.19e-01 | 7.58% |
| w8a8 | 42.41% | 0.02% | 0.17% | 4.63e-01 | 2.04e-01 | 2.74e-01 | 7.46% |
| KV-FP8 | 41.65% | 0.79% | 0.19% | **1.69e-05** | **9.28e-04** | **4.44e-04** | 9.03% |
| Baseline 3.3 70B | 57.15% | – | – | – | – | – | – |
| w4a16 | 17.70% | 39.46% | 0.43% | **0.00e+00** | **0.00e+00** | **0.00e+00** | 53.07% |
| FP8-dynamic | 56.96% | 0.20% | 0.13% | 6.33e-02 | 1.13e-01 | 6.06e-02 | 3.21% |
| w8a8 | 56.55% | 0.61% | 0.14% | **1.34e-05** | **4.00e-05** | **1.83e-04** | 4.18% |
| KV-FP8 | 56.87% | 0.29% | 0.14% | **2.51e-02** | **4.62e-02** | 7.57e-02 | 4.18% |
| Baseline Mistral | 56.74% | – | – | – | – | – | – |
| FP8-dynamic | 56.65% | 0.09% | 0.16% | 3.07e-01 | 5.69e-01 | 6.24e-01 | 6.83% |
| w4a16 | 55.87% | 0.87% | 0.21% | **1.30e-05** | **8.04e-03** | **2.84e-05** | 10.64% |
| Baseline 3.1B 8B | 32.72% | – | – | – | – | – | – |
| 2:4 Sparse | 30.13% | 2.59% | 0.29% | **1.09e-19** | **0.00e+00** | **1.89e-35** | 20.99% |

**Lossless variants:** For the Llama 3.1 8B Instruct model, we first run four *theoretically lossless* variants. For one, we simply rerun the exact same command for a second time, next we serve the model with the transformers engine (Wolf et al., 2020), then we serve the model on only a single GPU with tensor parallelism 1 (TP1). Lastly, we serve the model on a different hardware (A100) again using TP8. It is very important to note that even those, in theory, lossless changes lead to a notable number of flips of 1.3% to 2.8%. This fact is not noticed by Dutta et al. (2024) although it is known that changes in the computation graph can change the generation outputs.[6] We also observe that on some tasks the accuracy deviation can be over 0.5%, e.g., A100 on MUSR is 0.53% better than the baseline. A too tightly chosen hard threshold rule might already flag such deviations, though the two baseline rules ($\hat{\delta} > 2\%$ and $\hat{p}_{\updownarrow} \geq 5\%$) we consider do not flag any of those checkpoints. Similarly, our statistical framework clearly indicates that none of the models had significant degradation, in fact they all improve within expected statistical variations.

**Lossy variants:** To assess whether quantized models lead to *statistically significant* degradations we use a series of quantized model variants provided by RedHatAI (Kurtic et al., 2025), see Table 4. These checkpoints can readily be served with vLLM without any further flags. Furthermore, the last variant `KV-FP8` uses the original BF16 checkpoint with `kv_cache_dtype=fp8`, which uses FlashAttention3 in FP8 precision (Shah et al., 2024).

Our tests clearly flag all three INT4 checkpoints `w4a16` with extremely small $p$-values, indicating strong evidence that the model degraded significantly. In fact, on the w4a16 variant of the 70B model our evaluation indicates a bug in vLLM. We spot-checked the checkpoint in `transformers` (Wolf et al., 2020), where the accuracy drop is much less severe. On the 8B model also the `KV-FP8` variant

---

[6]vLLM for example does not guarantee stability of outcomes, see "Q: Can the output of a prompt vary across runs in vLLM?" https://docs.vllm.ai/en/v0.6.2/serving/faq.html.

gets clearly flagged by all tests. This is a particularly interesting case. Here, purely looking at the accuracy numbers without proper statistical tools would make a decision hard. On a few tasks, the model did not show degradation, while on MUSR it even improved slightly. However, our tests properly take sample size difference into account and the accuracy drops on the large datasets BBH, MATH, MMLU-pro are a strong signal of overall accuracy degradation. This is despite the pooled accuracy difference being only $0.78\%$, hence the baseline criterion $\hat{\delta} > 2\%$ would not detect it. We further illustrate this case in Figure 1. On the 70B model, we find that KV-FP8 has a drop of $0.3\%$ and is still detected with statistical significance, showing the power of our approach, while here both baseline criteria fail to detect the degradation.

We also note that within the Llama 8B Instruct experiments the two models that get clearly flagged have the highest flip probability, which seemingly aligns with the findings of Dutta et al. (2024). However, our analysis shows that the 8B FP8 has no accuracy concerns at all, while the flip probability ($8.71\%$) is almost the same as for 8B KV-FP8 ($9.03\%$) and thus would be flagged by the hard flip criterion. Based on the accuracy results, this might indeed be a false positive of the hard flip rule. Furthermore, on the 70B models the flip probability is generally lower and already quite close to the flip probabilities of the even theoretically lossless variants of the 8B model. We emphasize that on the 70B model none of the checkpoints would have been flagged by the baseline criteria. This underlines that a focus on a proper statistical treatment of accuracy is more appropriate than an investigation of the flip probability itself.

While some quantization approaches are indeed lossless, we also investigate a pruned and distilled variant of the Llama-3.1 8B Base model. This model has a large $2.6\%$ pooled accuracy drop and $p$-values that are effectively zero leaving no doubts about the statistical relevancy.

### 4.1 Reducing Dataset Sizes for Faster Degradation Testing

Recommendation 1 motivates that for a more effective degradation evaluation, we could consider trimming datasets to only include examples that are *likely* to flip. Dutta et al. (2024, Section 5) provided an analysis for classification tasks showing that examples are likely to flip if the probabilities of the top two answers are close, whereas we here aim to identify examples that flip in generative tasks. We use Llama-3.1 8B Instruct and the generative variant of MMLU-Pro (Wang et al., 2024) as an example because it is a very large dataset (12,032 examples) so here a reduction is most relevant to reduce the overall evaluation runtime of compressed models.

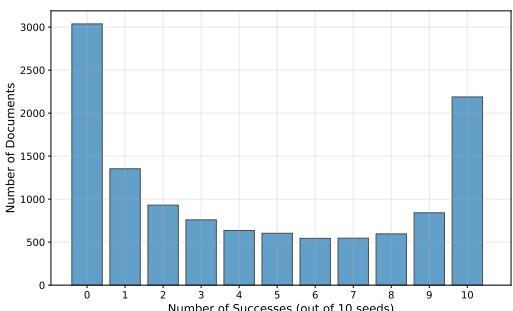

Figure 2: Success histogram for MMLU-Pro.

The intuition of our approach is that examples whose score flips when adding a bit of noise will also likely be the flips in actual degradation tests. On the other hand examples that are always correctly or always incorrectly solved even with noise are unlikely to flip in degradation tests. We thus run multiple iterations of the base model at finite temperature (10 iterations, Temp=0.3 chosen to lead to $\approx 20\%$ flips each time), whereas the task is usually evaluated at temperature 0. For each document in the dataset we then count how often it is solved correctly out of the 10 runs and plot this in a histogram, see Figure 2. It is quite striking that there are 5,604 examples in the dataset that never flip. We propose to remove those from the dataset when testing many different optimization approaches and only keep the other examples that are more likely to flip – and thus provide relevant information for degradation testing. To verify how relevant the identified examples from our simulation are for the real optimization degradation tests, we run KV-FP8 and w4a16 variants against the original model. Using all examples we obtain 2,472 / 12,032 ($20.55\%$) flips for w4a16 and 1,864 / 12,032 ($15.49\%$) for KV-FP8. After removing the 5,604 examples we identified to never flip in Figure 2, we obtain 2,116 / 6,807 ($31.09\%$) for w4a16 and 1,662 / 6,807 ($24.42\%$) for KV-FP8. Our approach thus allows us to almost halve the dataset size, while keeping most of the actually relevant examples in it. We believe that this is a promising direction to create compressed datasets in the future that are specifically designed for an efficient statistical evaluation of model degradation.

## 5 INTERPRETATION, EXTENSIONS AND DISCUSSION

**Interpretation of test outcomes:** Statistical hypothesis testing is a tool to facilitate binary decisions based on noisy observations. There will always be cases of type-I and type-II errors and we can only aim to minimize their rates. If the requirement is that an optimization is lossless and our tests flag it, we have clear and quantifiable statistical evidence that there is some accuracy degradation. In simplified terms, the $p$-value provides the probability of observing such an extreme empirical accuracy degradation if in fact the models are equally good. However, as is evident from the theory and also illustrated in Figure 5, there is no magical discontinuity at the test threshold, so even a checkpoint that barely passed might still have some (small) accuracy degradation. To provide a more comprehensive view, we derived the variance formula of the accuracy difference and report those in our code package and in Table 2. Furthermore, the usefulness of an optimized model is not solely dependent on its accuracy but also its expected performance gains. Thus, a checkpoint can still be useful despite having *some* accuracy loss.

**Extensions of our tests:** Some benchmarks use metrics that are non-binary, for example text summarization (Nallapati et al., 2016). For simplicity, we focused our main discussion on the binary case because of the special form of Equation (8). In Appendix D, we generalize our tests to non-binary metrics via a permutation approach. There we illustrate that the permutation-based test results in almost exactly the same $p$-values for binary metrics. On non-binary metrics, we could use the binary tests by thresholding scores, or adapting the contingency table to use $b$ and $c$ as win rates. However, this loses relevant information and our experiments in Appendix D.3 and Appendix D.4 show that our permutation approach is more effective. For non-binary scores we thus always recommend to use the tests of Appendix D.

Our tests being one-sided could also be considered as a limitation, however, it is a deliberate choice. Since the binomial distribution is symmetric for our null hypothesis ($q_\downarrow = 1/2$), we obtain 2x smaller $p$-values whenever there is an empirical accuracy degradation. This allows to be more sensitive when detecting degradation. If one were interested in general deviations, an adaption to a two-sided variant is straight-forward.

For simplicity, we focused our investigation on optimized model checkpoints. However, an equally important use-case for our proposed testing framework is integration testing in inference engines. For example, when developing a new kernel optimization or implementing an algorithm for speculative decoding, one can directly use the proposed tests to reliably assess the null hypothesis that the model did not accidentally degrade and thus to detect bugs in the implementation.

**Discussion:** Optimizing the performance of large LLMs is crucial and with that the assessment of the accuracy implications. Prior work has considered the empirical accuracy drop to qualitatively assess which quantization schemes are more favorable than others (Kurtic et al., 2025). Recent work by Dutta et al. (2024) has proposed to use the flip probability to assess whether a model is degraded or not. However, both approaches lack a proper treatment of the estimation uncertainty. In this work, we advocate that detecting model degradation should rely on the degradation probability rather than the flip probability alone. The flip probability rather indirectly affects our criteria by increasing the effective sample size ($b + c$) that our tests can use and in Section 4.1 we demonstrated how to adapt datasets to relatively increase the flips. Furthermore, as we demonstrate, flips happen naturally also in theoretically lossless optimizations like changes of the hardware or the serving configuration. The root cause is that modern LLMs and serving frameworks rely on 16-bit data types for activations, which can cause harmless numerical errors. Thus, there is no principled way to formulate a null hypothesis about flips, whereas for the degradation probability it is given by Fact 1.

The exact one-sided McNemar test is grounded in a statistical foundation and we demonstrated its usefulness for detecting model degradations in LLMs. However, such methods have yet to receive sufficient attention within the LLM research community that evaluates optimized models. Furthermore, the problem of aggregating outcomes from different benchmarks is not trivial and does not have a unique solution. Our three tests *pooled*, *max drop*, and *Fisher* are well motivated and cover different degradation scenarios as we illustrate in our synthetic experiments. Furthermore, on relevant LLMs they are powerful enough to detect even empirical accuracy degradations of 0.3% as statistically significant. We provide a script for the existing lm-eval package (Gao et al., 2024) to generate the statistical insights and analysis presented in this work. We anticipate that this will facilitate the future assessment of accuracy degradations both in industrial and academic contexts.

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

## A    NOTATION AND ALGORITHMS

In Table 3 we summarize the notation that we use throughout the paper and Algorithms 1 to 4 describe our (aggregation) algorithms.

Table 3: Notation Summary

| Symbol | Description | Formula |
|--------|-------------|---------|
| $M, \tilde{M}$ | Baseline and optimized models | – |
| $N$ | Total sample size | – |
| $T$ | Number of tasks | – |
| $a, b, c, d$ | Contingency table counts | – |
| $p_{\updownarrow}$ | Flip probability (models disagree) | $p_{\updownarrow} = \mathbb{E}[b+c]/N$ |
| $q_{\downarrow}$ | (Conditional) degradation probability | $q_{\downarrow} = P_b/p_{\updownarrow}$ |
| $\hat{q}_{\downarrow}$ | Empirical degradation probability | $\hat{q}_{\downarrow} = b/(b+c)$ |
| $\gamma, \beta$ | Baseline and optimized model accuracies | – |
| $\delta$ | Accuracy difference | $\delta = \gamma - \beta$ |
| $\hat{\delta}$ | Empirical accuracy difference | $\hat{\delta} = (b-c)/N$ |
| $P_b, P_c$ | Expected proportions | $P_b = p_{\updownarrow}q_{\downarrow}, P_c = p_{\updownarrow}(1 - q_{\downarrow})$ |
| $D(x)$ | Difference indicator function | See Equation (8) |
| $\alpha$ | Significance level | – |
| $\Phi$ | Standard normal CDF | – |
| $\Phi^{-1}$ | Standard normal quantile function | – |
| SNR | Signal-to-noise ratio | see Section 3.2 |
| $n_{\text{flips}}$ | Number of disagreements | $n_{\text{flips}} = b + c$ |
| $b_i, c_i$ | Task-specific counts | For task $i \in \{1, \ldots, T\}$ |
| $\hat{z}_i$ | Standardized degradation statistic | $\hat{z}_i = (\hat{q}_{\downarrow i} - 0.5)/\text{SE}(\hat{q}_{\downarrow i})$ |
| $\text{SE}(\hat{q}_{\downarrow i})$ | Standard error of degradation estimate | $\text{SE}(\hat{q}_{\downarrow i}) = \sqrt{0.25/(b_i + c_i)}$ |

---

**Algorithm 1** `McNemar`: Exact One-Sided McNemar Test

---

**Require:** Contingency table with counts $b, c$. Significance level $\alpha$.
1: $n_{flips} \leftarrow b + c$
2: **if** $n_{flips} = 0$ **then**
3:      **return** "No disagreements - Fail to reject $H_0$"
4: $\hat{q} \leftarrow \frac{b}{n_{flips}}$
5: **return** $pval \leftarrow$ `binomtest`$(k = b, n = n_{flips}, p = 0.5,$ alternative="greater").pvalue

---

**Algorithm 2** Pooled Test

---

**Require:** Lists $b_{list}, c_{list}$ for $T$ tasks. Significance level $\alpha$.
1: $b \leftarrow \sum_{i=1}^{T} b_{list}[i]$
2: $c \leftarrow \sum_{i=1}^{T} c_{list}[i]$
3: **return** `McNemar`$(b = b, c = c, \alpha)$

---

## B    VARIANCE OF THE ACCURACY DIFFERENCE INDICATOR

We here derive the variance of the accuracy difference indicator function defined as

$$D(x) = \begin{cases} 0 & \text{if } L(M(x)) = L(\tilde{M}(x)), \\ 1 & \text{if } L(M(x)) = 1 \text{ and } L(\tilde{M}(x)) = 0, \\ -1 & \text{if } L(M(x)) = 0 \text{ and } L(\tilde{M}(x)) = 1. \end{cases} \tag{13}$$

Denoting the population probabilities with $P_a, P_b, P_c, P_d$ we have

$$\mathbb{E}[D(X)] = P_b - P_c, \tag{14}$$

$$\mathbb{E}[D^2(X)] = P_b + P_c. \tag{15}$$

---

**Algorithm 3** Fisher Aggregation Test

---
**Require:** Lists $b_{list}$, $c_{list}$ for $T$ tasks. Significance level $\alpha$.
1: **for** $i = 1$ to $T$ **do**
2:     $p_i \leftarrow$ McNemar$(b = b_{list}[i], c = c_{list}[i], \alpha)$
3: $\chi^2_{stat} \leftarrow -2 \sum_{i=1}^{T} \ln(p_i)$
4: **return** $pval \leftarrow$ chi2.sf$(\chi^2_{stat}, 2T)$

---

**Algorithm 4** Max Drop Test

---
**Require:** Lists $b_{list}$, $c_{list}$ for $T$ tasks. Simulation steps $S$.
1: $n_{list} \leftarrow [b_{list}[i] + c_{list}[i]]_{i=1}^{T}$
2: **for** $s = 1$ to $S$ **do**
3:     **for** $i = 1$ to $T$ **do**
4:         $b_{sim}[s, i] \leftarrow$ binomial$(n_{list}[i], 0.5)$
5:         $\hat{q}_{sim}[s, i] \leftarrow \frac{b_{sim}[s,i]}{n_{list}[i]} - 0.5$
6:         $\sigma[s, i] \leftarrow \sqrt{\frac{0.25}{n_{list}[i]}}$
7:     $z_{sim}[s] \leftarrow \max_i \frac{\hat{q}_{sim}[s,i]}{\sigma[s,i]}$
8: $\hat{q}_{obs}[i] \leftarrow \frac{b_{list}[i]}{n_{list}[i]} - 0.5$, $\sigma_{obs}[i] \leftarrow \sqrt{\frac{0.25}{n_{list}[i]}}$
9: $z_{obs} \leftarrow \max_i \frac{\hat{q}_{obs}[i]}{\sigma_{obs}[i]}$
10: **return** $pval \leftarrow \frac{1}{S} \sum_{s=1}^{S} \mathbf{1}[z_{sim}[s] \geq z_{obs}]$

---

And we obtain

$$\text{Var}[D(X)] = \mathbb{E}[D^2(X)] - \mathbb{E}[D(X)]^2 \tag{16}$$
$$= P_b + P_c - (P_b - P_c)^2 \tag{17}$$

## C  FURTHER EXPERIMENTS AND DETAILS

### C.1  LLM EXPERIMENTS

As detailed in the main paper, our default for running the experiments was to use tensor parallelism of degree 8. However, for the A100 and the KV8 experiments, we had to run MATH on a single device, i.e., TP1 as otherwise we encountered a bug in the collectives in vLLM.

In Figure 5 we additionally illustrate how the p-values of the pooled test depends on the overall sample size $N$, the empirical accuracy degradation $\hat{\delta}$ as well as the flip probability $p_{\updownarrow}$.

### C.2  SYNTHETIC EXPERIMENTS

To gain further insights into the properties of the proposed aggregation strategies, we evaluate the statistical tests across three scenarios with varying numbers of tasks $T \in \{1, 2, \ldots, 20\}$. For each task, we simulate $N$ samples where $N \sim$ Uniform$(500, 10000)$, with flip probability $p = 0.1$ across all scenarios. We simulate 1000 experiments per configuration and test at significance level $\alpha = 0.05$.

**Scenario 1 (Type-I Error):** Under the null hypothesis, we set degradation probability $q_{\downarrow} = 0.5$ for all tasks to measure false positive rates.

**Scenario 2 (Balanced Degradation):** Under the alternative hypothesis, we set $q = 0.52$ uniformly across all tasks to simulate consistent model degradation.

**Scenario 3 (Single Task Degradation):** We set $q = 0.58$ for the first task only and $q = 0.5$ for all remaining tasks to simulate degradation for a single specific task.

For each scenario, we report rejection rates at the 5% significance level in Figure 3: type-I error rates for Scenario 1, and test power (1 - type-II error rate) for Scenarios 2 and 3. First we observe

Table 4: Full model checkpoint specifiers used in the experiments. We refer to the HuggingFace model cards for full information about their configurations.

| ID | HuggingFace Model Repository |
|---|---|
| Llama-3.1 8B | meta-llama/Llama-3.1-8B-Instruct |
| w4a16 | RedHatAI/Meta-Llama-3.1-8B-Instruct-quantized.w4a16 |
| FP8 | RedHatAI/Meta-Llama-3.1-8B-Instruct-FP8 |
| w8a16 | RedHatAI/Meta-Llama-3.1-8B-Instruct-quantized.w8a16 |
| FP8-dynamic | RedHatAI/Meta-Llama-3.1-8B-Instruct-FP8-dynamic |
| w8a8 | RedHatAI/Meta-Llama-3.1-8B-Instruct-quantized.w8a8 |
| KV-FP8 | meta-llama/Llama-3.1-8B-Instruct vLLM flag `kv_cache_dtype=fp8` |
| Llama 3.3 70B Instruct | meta-llama/Llama-3.3-70B-Instruct |
| w4a16 | RedHatAI/Llama-3.3-70B-Instruct-quantized.w4a16 |
| FP8-dynamic | RedHatAI/Llama-3.3-70B-Instruct-FP8-dynamic |
| w8a8 | RedHatAI/Llama-3.3-70B-Instruct-quantized.w8a8 |
| KV-FP8 | meta-llama/Llama-3.3-70B-Instruct + vLLM flag `kv_cache_dtype=fp8` |
| Llama 3.1B 8B | meta-llama/Llama-3.1-8B |
| 2:4 Sparse | RedHatAI/Sparse-Llama-3.1-8B-2of4 |
| Mistral | mistralai/Mistral-Small-3.1-24B-Instruct-2503 |
| FP8-dynamic | RedHatAI/Mistral-Small-3.1-24B-Instruct-2503-FP8-dynamic |
| w4a16 | RedHatAI/Mistral-Small-3.1-24B-Instruct-2503-quantized.w4a16 |
| gpt-oss 20B | openai/gpt-oss-20b |
| KV-FP8 | openai/gpt-oss-20b vLLM flag `kv_cache_dtype=fp8` |

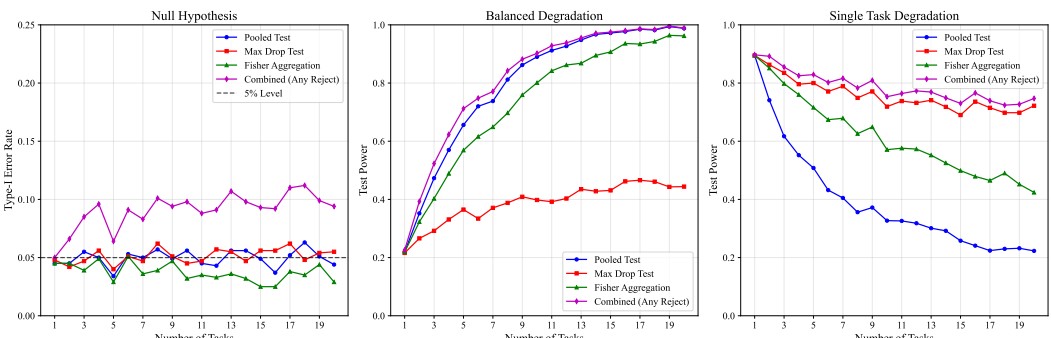

Figure 3: Rejection rates of the proposed aggregation schemes dependent on the number of tasks $T$. Error bars are left implicit and are given by the variance formula $\text{Var}[p] = p(1-p)/1000$, since we estimate each probability from 1000 synthetic experiments.

that all our tests correctly control (up to finite sample deviations) the type-I error at or below the significance level $\alpha = 5\%$. For larger $T$ the Fisher aggregation is slightly more conservative, which can be explained by the fact that the binomial distribution is discrete. Thus the individual $p$-values are not exactly uniformly distributed but slightly more conservative, an effect that disappears when each task has a large number of flips (Lancaster, 1961).

For the test power scenarios, as expected, the pooled test yields the highest rejection rate when all tasks are equally affected by degradations, whereas the Max Drop works best on single task degradation. For single task degradation, the experiment also nicely illustrates the downsides of adding data(sets) that do only add noise but no signal. In both simulated scenarios Fisher strikes a good balance in between the two extremes.

Table 5: Accuracy results of optimized Llama-3.1 8B Instruct, 3.3 70B Instruct, Mistral Small, and Llama-3.1 8B (non-instruct) variants.

| Model
# Samples → | BBH
(5761) | GPQA
(1192) | IFEVAL
(541) | MATH
(5000) | MMLU-Pro
(12032) | MUSR
(756) |
|---|---|---|---|---|---|---|
| Baseline 3.1 8B | 50.53% | 28.61% | 74.86% | 45.22% | 37.56% | 38.49% |
| Baseline (rerun) | 50.53% | 28.61% | 75.05% | 45.30% | 37.56% | 38.49% |
| transformers | 50.69% | 28.52% | 73.94% | 45.18% | 37.77% | 38.49% |
| TP1 | 50.62% | 28.69% | 74.49% | 44.86% | 37.68% | 38.76% |
| A100 | 50.62% | 28.61% | 74.86% | 45.26% | 37.77% | 39.02% |
| w4a16 | 50.10% | 28.02% | 70.98% | 41.18% | 36.06% | 38.10% |
| FP8 | 50.79% | 29.78% | 73.94% | 44.28% | 37.86% | 38.23% |
| w8a16 | 50.60% | 27.52% | 74.12% | 44.56% | 37.77% | 37.57% |
| FP8-dynamic | 50.06% | 28.61% | 76.16% | 44.72% | 37.84% | 38.36% |
| w8a8 | 50.84% | 28.36% | 75.97% | 44.36% | 37.78% | 37.17% |
| KV-FP8 | 50.25% | 28.61% | 74.86% | 43.20% | 36.84% | 39.02% |
| Baseline 3.3 70B | 69.02% | 31.38% | 89.09% | – | 53.41% | 44.18% |
| w4a16 | 27.58% | 26.51% | 12.75% | – | 11.17% | 35.85% |
| FP8-dynamic | 68.91% | 30.62% | 88.54% | – | 53.26% | 43.65% |
| w8a8 | 68.63% | 30.96% | 89.09% | – | 52.66% | 43.39% |
| KV-FP8 | 69.05% | 30.70% | 89.65% | – | 52.94% | 44.31% |
| Baseline Mistral | 66.98% | 39.85% | 63.96% | 52.80% | 55.45% | 46.56% |
| FP8-dynamic | 67.14% | 39.77% | 63.40% | 52.62% | 55.26% | 47.22% |
| w4a16 | 66.50% | 38.26% | 58.41% | 51.34% | 54.85% | 46.96% |
| Baseline 3.1B 8B | 46.42% | 29.61% | 7.95% | 19.02% | 32.92% | 38.49% |
| 2:4 Sparse | 46.50% | 28.52% | 13.31% | 12.70% | 29.41% | 46.56% |

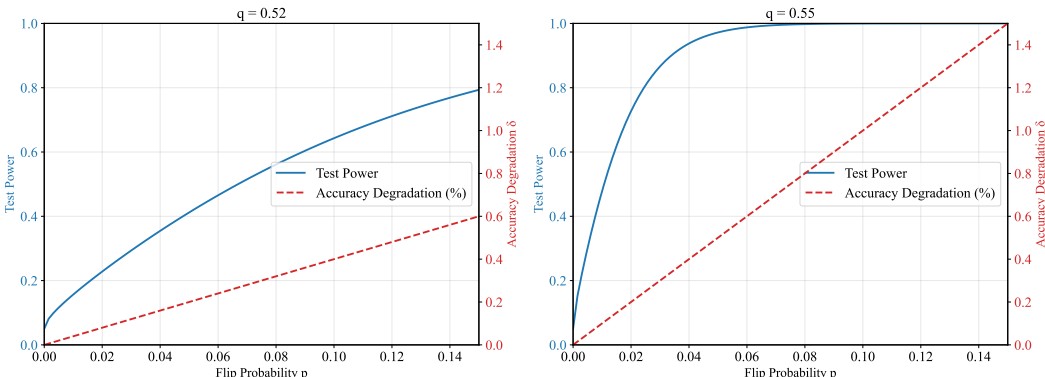

Figure 4: Asymptotic test power for $N = 25,282$ and $\alpha = 0.05$ as function of flip probability $p_{\updownarrow}$ and degradation probability $q_{\downarrow}$.

We also additionally show the rejection rates of an aggregated test that rejects if any of the three tests rejects at $\alpha = 5\%$. This test of course does not control correctly at the significance level, however, it captures all potential scenarios.

Table 6: Full contingency table for Llama-3.1 8B Instruct experiments. $a, b, c, d$ are as defined in Table 1 and $p_{\updownarrow}$ denotes the flip probability (cf. Section 3).

| Model | | BBH | GPQA | IFEVAL | MATH | MMLU | MUSR | Total |
|---|---|---|---|---|---|---|---|---|
| Baseline (rerun) | b | 0 | 0 | 1 | 162 | 0 | 0 | 163 |
| Baseline (rerun) | c | 0 | 0 | 2 | 166 | 0 | 0 | 168 |
| Baseline (rerun) | d | 2911 | 341 | 404 | 2099 | 4519 | 291 | 10565 |
| Baseline (rerun) | $p_{\updownarrow}$ | 0.00% | 0.00% | 0.55% | 6.56% | 0.00% | 0.00% | 1.31% |
| TP1 | a | 2812 | 847 | 119 | 2579 | 7430 | 462 | 14249 |
| TP1 | b | 33 | 3 | 19 | 178 | 68 | 1 | 302 |
| TP1 | c | 38 | 4 | 17 | 160 | 83 | 3 | 305 |
| TP1 | d | 2878 | 338 | 386 | 2083 | 4451 | 290 | 10426 |
| TP1 | $p_{\updownarrow}$ | 1.23% | 0.59% | 6.65% | 6.76% | 1.25% | 0.53% | 2.40% |
| A100 | a | 2810 | 838 | 119 | 2554 | 7409 | 461 | 14191 |
| A100 | b | 35 | 13 | 17 | 183 | 79 | 0 | 327 |
| A100 | c | 40 | 13 | 17 | 185 | 104 | 4 | 363 |
| A100 | d | 2876 | 328 | 388 | 2078 | 4440 | 291 | 10401 |
| A100 | $p_{\updownarrow}$ | 1.30% | 2.18% | 6.28% | 7.36% | 1.52% | 0.53% | 2.73% |
| w4a16 | a | 2480 | 809 | 107 | 2368 | 6981 | 431 | 13176 |
| w4a16 | b | 395 | 49 | 50 | 573 | 712 | 37 | 1816 |
| w4a16 | c | 370 | 42 | 29 | 371 | 532 | 34 | 1378 |
| w4a16 | d | 2516 | 292 | 355 | 1688 | 3807 | 254 | 8912 |
| w4a16 | $p_{\updownarrow}$ | 13.28% | 7.63% | 14.60% | 18.88% | 10.34% | 9.39% | 12.63% |
| FP8 | a | 2646 | 812 | 109 | 2362 | 7070 | 449 | 13448 |
| FP8 | b | 189 | 25 | 32 | 424 | 407 | 18 | 1095 |
| FP8 | c | 204 | 39 | 27 | 377 | 443 | 16 | 1106 |
| FP8 | d | 2722 | 316 | 373 | 1837 | 4112 | 273 | 9633 |
| FP8 | $p_{\updownarrow}$ | 6.82% | 5.37% | 10.91% | 16.02% | 7.06% | 4.50% | 8.71% |
| w8a16 | a | 2767 | 847 | 110 | 2469 | 7350 | 462 | 14005 |
| w8a16 | b | 79 | 17 | 30 | 303 | 138 | 10 | 577 |
| w8a16 | c | 83 | 4 | 26 | 270 | 163 | 3 | 549 |
| w8a16 | d | 2832 | 324 | 375 | 1958 | 4381 | 281 | 10151 |
| w8a16 | $p_{\updownarrow}$ | 2.81% | 1.76% | 10.35% | 11.46% | 2.50% | 1.72% | 4.45% |
| FP8-dynamic | a | 2670 | 830 | 106 | 2385 | 7157 | 454 | 13602 |
| FP8-dynamic | b | 207 | 21 | 23 | 379 | 322 | 12 | 964 |
| FP8-dynamic | c | 180 | 21 | 30 | 354 | 356 | 11 | 952 |
| FP8-dynamic | d | 2704 | 320 | 382 | 1882 | 4197 | 279 | 9764 |
| FP8-dynamic | $p_{\updownarrow}$ | 6.72% | 3.52% | 9.80% | 14.66% | 5.63% | 3.04% | 7.58% |
| w8a8 | a | 2674 | 827 | 104 | 2382 | 7171 | 455 | 13613 |
| w8a8 | b | 158 | 27 | 26 | 400 | 315 | 20 | 946 |
| w8a8 | c | 176 | 24 | 32 | 357 | 342 | 10 | 941 |
| w8a8 | d | 2753 | 314 | 379 | 1861 | 4204 | 271 | 9782 |
| w8a8 | $p_{\updownarrow}$ | 5.80% | 4.28% | 10.72% | 15.14% | 5.46% | 3.97% | 7.46% |
| KV-FP8 | a | 2616 | 826 | 113 | 2392 | 7120 | 445 | 13512 |
| KV-FP8 | b | 250 | 25 | 23 | 448 | 479 | 16 | 1241 |
| KV-FP8 | c | 234 | 25 | 23 | 347 | 393 | 20 | 1042 |
| KV-FP8 | d | 2661 | 316 | 382 | 1813 | 4040 | 275 | 9487 |
| KV-FP8 | $p_{\updownarrow}$ | 8.40% | 4.19% | 8.50% | 15.90% | 7.25% | 4.76% | 9.03% |

Table 7: Full contingency table for Llama-3.3 70B Instruct experiments. $a, b, c, d$ are as defined in Table 1 and $p_\updownarrow$ denotes the flip probability (cf. Section 3).

| Model | | BBH | GPQA | IFEVAL | MMLU | MUSR | Total |
|---|---|---|---|---|---|---|---|
| w4a16 | a | 1342 | 595 | 53 | 5002 | 318 | 7310 |
| w4a16 | b | 2830 | 281 | 419 | 5686 | 167 | 9383 |
| w4a16 | c | 443 | 223 | 6 | 604 | 104 | 1380 |
| w4a16 | d | 1146 | 93 | 63 | 740 | 167 | 2209 |
| w4a16 | $p_\updownarrow$ | 56.81% | 42.28% | 78.56% | 52.28% | 35.85% | 53.07% |
| FP8-dynamic | a | 1708 | 813 | 51 | 5394 | 418 | 8384 |
| FP8-dynamic | b | 83 | 14 | 11 | 230 | 8 | 346 |
| FP8-dynamic | c | 77 | 5 | 8 | 212 | 4 | 306 |
| FP8-dynamic | d | 3893 | 360 | 471 | 6196 | 326 | 11246 |
| FP8-dynamic | $p_\updownarrow$ | 2.78% | 1.59% | 3.51% | 3.67% | 1.59% | 3.21% |
| w8a8 | a | 1656 | 810 | 48 | 5396 | 418 | 8328 |
| w8a8 | b | 151 | 13 | 11 | 300 | 10 | 485 |
| w8a8 | c | 129 | 8 | 11 | 210 | 4 | 362 |
| w8a8 | d | 3825 | 361 | 471 | 6126 | 324 | 11107 |
| w8a8 | $p_\updownarrow$ | 4.86% | 1.76% | 4.07% | 4.24% | 1.85% | 4.18% |
| KV-FP8 | a | 1665 | 812 | 51 | 5352 | 415 | 8295 |
| KV-FP8 | b | 118 | 14 | 5 | 310 | 6 | 453 |
| KV-FP8 | c | 120 | 6 | 8 | 254 | 7 | 395 |
| KV-FP8 | d | 3858 | 360 | 477 | 6116 | 328 | 11139 |
| KV-FP8 | $p_\updownarrow$ | 4.13% | 1.68% | 2.40% | 4.69% | 1.72% | 4.18% |

Table 8: Full contingency table for Mistral Small experiments. $a, b, c, d$ are as defined in Table 1 and $p_\updownarrow$ denotes the flip probability (cf. Section 3).

| Model | | BBH | GPQA | IFEVAL | MATH | MMLU | MUSR | Total |
|---|---|---|---|---|---|---|---|---|
| FP8-dynamic | a | 1793 | 670 | 163 | 1912 | 5164 | 383 | 10085 |
| FP8-dynamic | b | 100 | 48 | 35 | 457 | 219 | 16 | 875 |
| FP8-dynamic | c | 109 | 47 | 32 | 448 | 196 | 21 | 853 |
| FP8-dynamic | d | 3759 | 427 | 311 | 2183 | 6453 | 336 | 13469 |
| FP8-dynamic | $p_\updownarrow$ | 3.63% | 7.97% | 12.38% | 18.10% | 3.45% | 4.89% | 6.83% |
| w4a16 | a | 1711 | 668 | 161 | 1856 | 4942 | 365 | 9703 |
| w4a16 | b | 219 | 68 | 64 | 577 | 490 | 36 | 1454 |
| w4a16 | c | 191 | 49 | 34 | 504 | 418 | 39 | 1235 |
| w4a16 | d | 3640 | 407 | 282 | 2063 | 6182 | 316 | 12890 |
| w4a16 | $p_\updownarrow$ | 7.12% | 9.82% | 18.11% | 21.62% | 7.55% | 9.92% | 10.64% |

Table 9: Full contingency table for Llama-3.1 8B Base experiments. $a, b, c, d$ are as defined in Table 1 and $p_\updownarrow$ denotes the flip probability (cf. Section 3).

| Model | | BBH | GPQA | IFEVAL | MATH | MMLU | MUSR | Total |
|---|---|---|---|---|---|---|---|---|
| 2:4 Sparse | a | 2446 | 657 | 442 | 3733 | 7065 | 341 | 14684 |
| 2:4 Sparse | b | 636 | 195 | 27 | 632 | 1428 | 63 | 2981 |
| 2:4 Sparse | c | 641 | 182 | 56 | 316 | 1006 | 124 | 2325 |
| 2:4 Sparse | d | 2038 | 158 | 16 | 319 | 2533 | 228 | 5292 |
| 2:4 Sparse | $p_\updownarrow$ | 22.17% | 31.63% | 15.34% | 18.96% | 20.23% | 24.74% | 20.99% |

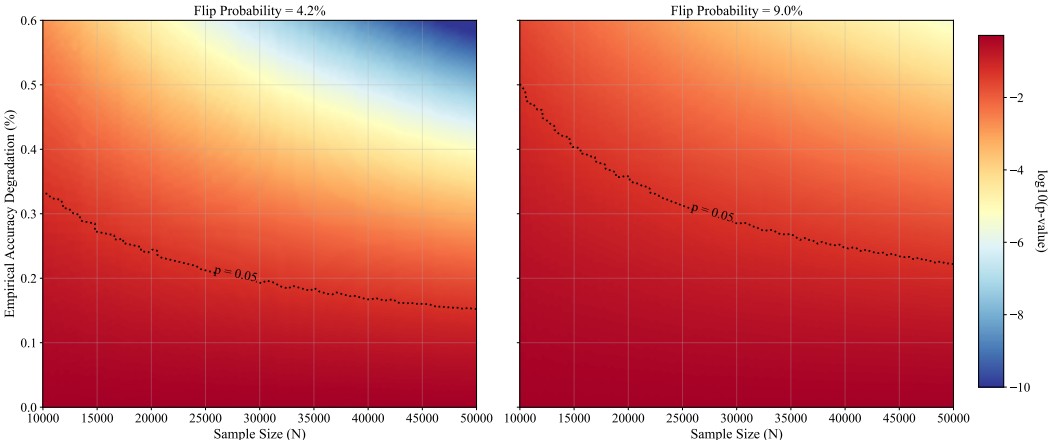

Figure 5: $p$-values of the pooled test as a function of sample size $N$, empirical accuracy degradation $\hat{\delta}$ and different flip probabilities. Observations above the dotted line are rejected as significant degradations at $\alpha = 5\%$.

## D  GENERALIZATION TO NON-BINARY LOSS

For ease of presentation in the main paper we focused on binary scores, where we can model everything via binomial distributions. A naive way to accommodate non-binary losses is to either threshold them to binary scores or to simply populate the contingency table by counting how often the $M$ scores higher than $\tilde{M}$ and vice-versa, and just ignoring cases where they have the same score.

However, for any reasonable scoring criterion, we should assume that a score of 0.9 versus 0.1 carries more information about a degradation than 0.6 versus 0.59, for example. In the following we motivate that this is also the right approach to take when evaluating models multiple times on the same input with finite temperature (Appendix D.1) and then introduce a generalized test based on permutations in Appendix D.2.

### D.1  MULTIPLE RUNS OF THE SAME BENCHMARK

Many recent LLMs are *reasoning* models and use longer token sequences to reason about a task before providing a final answer (Guo et al., 2025; Agarwal et al., 2025; Yang et al., 2025a). To achieve highest accuracy, such models are usually used at non-zero temperature. This brings additional randomness into the evaluation process, as an input example will now generally have a certain chance of being correct or false. To reduce this additional noise such models are often evaluated multiple times, especially for tasks with small datasets. For example, as of fall 2025, the Artificial Analysis Intelligence Index runs the AIME25 task, which only has 30 datapoints, ten times. Whereas they evaluate MMLU Pro, which has over 12,000 examples, only a single time.[7] Similarly, the GPT-OSS evals package, by default runs benchmarks multiple times.[8]

If we were to use our binary framework and just treat repetitions of the same samples as independent runs we would obtain unreliable $p$-values. To see this, consider the limiting case where the reruns are all done at temperature 0, or an arbitrary small finite temperature. And let us therefore assume that all $R \in \mathbb{N}$ reruns lead to the same contingency table values $b$ and $c$. Then adding more reruns clearly does not increase the statistical significance of the results. Yet, if $b > c$, the $p$-value of the binomial test can get arbitrarily small when $R \to \infty$.

Instead, we need to aggregate the score on the sample level, which we do by generalizing our scoring criterion to continuous values. For an example $x_i$, the score is now the expectation over infinite repetitions, i.e., $L_M(x_i) = \lim_{R \to \infty} \frac{1}{R} \sum_{r=1}^{R} L(M(x_i, r))$, where $r$ enumerates the repetitions and we leave the temperature settings implicit. For $R \in \mathbb{N}$, for a given task we thus in the end have a list of finite-repetition estimates $[\hat{L}_M(x_1), \dots, \hat{L}_M(x_N)]$, where $\hat{L}_M(x_i) = \frac{1}{R} \sum_{r=1}^{R} L(M(x_i, r))$.

### D.2  PERMUTATION-BASED CRITERIA FOR NON-BINARY SCORES

For continuous scores, we cannot use tests based on the binomial distribution anymore. For large datasets, we could use tests based on the asymptotic normality as discussed in Section 3.2. However, since some datasets are small, this would require additional care to check whether the asymptotic approximations are usable. We instead rely on a permutation-based test that guarantees type-I error control for any sample size. Under the null hypothesis, both models should perform equally well and thus any test statistic we compute based on $[\hat{L}_M(x_1), \dots, \hat{L}_M(x_N)]$ and $[\hat{L}_{\tilde{M}}(x_1), \dots, \hat{L}_{\tilde{M}}(x_N)]$ should not be systematically larger than if we computed it on two lists where we randomly permute the $i$-th entry of the list for all $i$. We generally draw $m \in \mathbb{N}$ permutations (with replacement) and count with $b \in \mathbb{N}$ the cases where the simulated test statistic is larger than the original one. Then the $p$-value $p = \frac{b+1}{m+1}$ always correctly controls type-I errors (Phipson & Smyth, 2010).

We implement three permutation-based variants of our tests. The *Permutation Pooled Test* (Algorithm 5) directly extends the pooled approach by computing the mean difference across all examples and comparing against a null distribution generated by random sign flips. The *Permutation Fisher Test* (Algorithm 6) applies permutation tests to each task individually, then combines p-values using Fisher's method. The *Permutation Max Drop Test* (Algorithm 7) standardizes per-task differences

---

[7]https://artificialanalysis.ai/methodology/intelligence-benchmarking#intelligence-index-evaluation-suite-summary
[8]https://github.com/openai/gpt-oss/tree/main/gpt_oss/evals

by their standard errors and uses the maximum standardized difference as the test statistic, analogous to our binary max drop test but adapted for continuous scores.

---

**Algorithm 5** Permutation Pooled Test

---

**Require:** Score lists $L_M = [\hat{L}_M(x_1), \ldots, \hat{L}_M(x_N)]$, $L_{\tilde{M}} = [\hat{L}_{\tilde{M}}(x_1), \ldots, \hat{L}_{\tilde{M}}(x_N)]$ for all tasks combined. Permutations $m$.
1: $\Delta \leftarrow [\hat{L}_M(x_i) - \hat{L}_{\tilde{M}}(x_i)]_{i=1}^N$
2: $d_{obs} \leftarrow \frac{1}{N} \sum_{i=1}^N \Delta[i]$
3: $b \leftarrow 0$
4: **for** $s = 1$ to $m$ **do**
5: $\quad signs \leftarrow \texttt{random\_choice}(\{-1, 1\}, \text{size}=N)$
6: $\quad d_{sim} \leftarrow \frac{1}{N} \sum_{i=1}^N signs[i] \cdot \Delta[i]$
7: $\quad$ **if** $d_{sim} \geq d_{obs}$ **then**
8: $\quad\quad b \leftarrow b + 1$
9: **return** $pval \leftarrow \frac{b+1}{m+1}$

---

**Algorithm 6** Permutation Fisher Aggregation Test

---

**Require:** Score lists $L_{M,t}$, $L_{\tilde{M},t}$ for $T$ tasks. Permutations $m$.
1: **for** $t = 1$ to $T$ **do**
2: $\quad p_t \leftarrow \texttt{PermutationPooled}(L_{M,t}, L_{\tilde{M},t}, m)$
3: $\chi^2_{stat} \leftarrow -2 \sum_{t=1}^T \ln(p_t)$
4: **return** $pval \leftarrow \texttt{chi2.sf}(\chi^2_{stat}, 2T)$

---

**Algorithm 7** Permutation Max Drop Test

---

**Require:** Score lists $L_{M,t}$, $L_{\tilde{M},t}$ for $T$ tasks. Permutations $m$. Epsilon $\epsilon = 10^{-10}$.
1: **for** $t = 1$ to $T$ **do**
2: $\quad \Delta_t \leftarrow [\hat{L}_{M,t}(x_i) - \hat{L}_{\tilde{M},t}(x_i)]_{i=1}^{N_t}$
3: $\quad d_{obs}[t] \leftarrow \frac{1}{N_t} \sum_{i=1}^{N_t} \Delta_t[i]$
4: $\quad \sigma[t] \leftarrow \frac{\texttt{std}(\Delta_t, \text{ddof}=1)}{\sqrt{N_t}} + \epsilon$
5: $z_{obs} \leftarrow \max_t \frac{d_{obs}[t]}{\sigma[t]}$
6: $b \leftarrow 0$
7: **for** $s = 1$ to $m$ **do**
8: $\quad$ **for** $t = 1$ to $T$ **do**
9: $\quad\quad signs_t \leftarrow \texttt{random\_choice}(\{-1, 1\}, \text{size}=N_t)$
10: $\quad\quad d_{sim}[t] \leftarrow \frac{1}{N_t} \sum_{i=1}^{N_t} signs_t[i] \cdot \Delta_t[i]$
11: $\quad z_{sim} \leftarrow \max_t \frac{d_{sim}[t]}{\sigma[t]}$
12: $\quad$ **if** $z_{sim} \geq z_{obs}$ **then**
13: $\quad\quad b \leftarrow b + 1$
14: **return** $pval \leftarrow \frac{b+1}{m+1}$

---

### D.3 Synthetic Experiments

We claimed that the permutation-based test should a/ give very similar $p$-values when used on binary scores and should better detect small differences when used with continuous scores. We first illustrate this with two synthetic examples. As we show in Table 10, the p-values are nearly identical (maximum absolute difference of 0.000919), confirming that permutation tests provide equivalent results to the exact binomial approach on binary data.

For the second test, we generate two tasks with a small accuracy degradation and continuous scores. To evaluate the binary tests here, we populate the contingency table based on whether the score between the models is larger or less. The alternative would be to threshold, but this is even worse.

Table 10: Comparison of p-values: Binary tests vs Permutation tests on binary data

| Test Method | Pooled | Max Drop | Fisher |
|---|---|---|---|
| Binary (McNemar-based) | 0.220127 | 0.474152 | 0.318846 |
| Permutation | 0.219208 | 0.474105 | 0.318749 |
| Absolute Difference | 0.000919 | 0.000047 | 0.000097 |

As shown in Table 11, the permutation-based tests are substantially more sensitive, detecting the degradation with p-values that are 2-10 times smaller than the binary approach. This demonstrates that thresholding continuous scores to binary loses important statistical power for detecting subtle performance differences.

Table 11: Comparison of p-values: Binary tests vs Permutation tests on continuous data with accuracy degradation

| Test Method | Pooled | Max Drop | Fisher |
|---|---|---|---|
| Binary (thresholded) | 0.073483 | 0.056188 | 0.070337 |
| Permutation | 0.007230 | 0.024320 | 0.011628 |
| Ratio (Binary/Permutation) | 10.16 | 2.31 | 6.05 |

### D.4 LLM EXPERIMENTS

**Llama 3.1 8B Instruct on TruthfulQA**    To strengthen our claim that a reduction to a binary test sacrifices test power, we evaluate two lossy Llama-3.1 8B Instruct variants on the task TruthfulQA (Lin et al., 2022). We run the generative variant of lm-eval Gao et al. (2024) and use the metric RougeL Max, which measures the RougeL score of the LLMs generation with all options for correct answers and then considers the maximum as final score. This metric is inherently continuous. We then compute the (single task) pooled $p$-value with a/ the permutation-based approach (recommended), b/ our binary script where we threshold the scores to 0 or one at 0.5, and c/ where we consider just the win/loss rates for the contingency table.

The results in Table 12 confirm that the permutation-based test is the most sensitive test and should always be used for non-binary scores.

Table 12: Comparison of $p$-values: Binary tests vs Permutation tests on continuous data from TruthfulQA on two variants of Llama-3.1 8B Instruct.

| Model | Average RougeLMax | Permutation | Threshold | Win-Rate |
|---|---|---|---|---|
| Baseline | 59.22% | – | – | – |
| w4a16 | 55.26% | 1.00e-5 | 0.042 | 0.023 |
| w816 | 58.21% | 0.0014 | 0.047 | 0.0018 |

**GPT-OSS**    For GPT-OSS models (Agarwal et al., 2025), OpenAI released their own evaluation suite[9]. We can use this to evaluate the 20B model on GPQA (Rein et al., 2023) and AIME25[10]. Since the GPT-OSS models already come with their MoE modules in MX-FP4 precision by default, we could not find meaningful models that are further quantized. We therefore compare the 20B model against a rerun and against a version with FP8 KV cache. We also include a pruned variant which only has 7 experts per MoE layer `gpt-oss-6.0b-specialized-all-pruned-moe-only-7-experts`[11]. However, this turns out to be way too aggressive. Furthermore, to have a model with more gradual degradation,

---

[9]`https://github.com/openai/gpt-oss/tree/main/gpt_oss/evals`
[10]https://huggingface.co/datasets/opencompass/AIME2025
[11]https://huggingface.co/AmanPriyanshu/gpt-oss-6.0b-specialized-all-pruned-moe-only-7-experts

we also include a run for which as "optimization" we constrain the models context to 32,000 instead of 128,000 tokens.

The peculiarity of the GPT-OSS eval suite is that each example of the dataset is evaluated with three different levels of reasoning effort and each of them is run eight times with finite temperature. Thus, although each individual run results in a binary score, when aggregating over the 24 runs each example is evaluated on, we get a more fine-grained score. Thus, we also need to apply the permutation-based tests to obtain the strictest $p$-values as outlined above. The results of the tests are presented in Table 13. While the rerun and FP8 KV cache are effectively lossless, the pruned model and the model that is artificially constrained to 32,000 context length show clear accuracy drops that are detected as statistically significant. Notice that for our experiments we use $m = 10^5$ permutations and thus the smallest possible $p$-values for the pooled test and the max drop test are $1/(10^5 + 1) \approx 1.00\text{e-}05$. If one wishes to obtain even more sensitive $p$-values, one can further increase the number of permutations, but we do not foresee practical relevance for smaller $p$-values.

Table 13: Evaluation of variants of GPT-OSS 20B with the OpenAI evaluation suite.

| model | $\hat{\gamma}$ | $\hat{\delta}$ | $p_{\text{pool}}$ | $p_{\text{fisher}}$ | $p_{\text{max drop}}$ | AIME25 | GPQA |
|---|---|---|---|---|---|---|---|
| Baseline 20B | 65.59% | – | – | – | – | 64.44% | 65.76% |
| Baseline (rerun) | 66.06% | -0.47% | 7.59e-01 | 8.38e-01 | 9.07e-01 | 65.41% | 66.16% |
| KV-FP8 | 65.42% | 0.16% | 3.87e-01 | 5.54e-01 | 5.75e-01 | 64.99% | 65.49% |
| pruned 7 exp | 21.91% | 43.7% | **1.00e-05** | **2.40e-09** | **1.00e-05** | 0.14% | 25.21% |
| max32k | 58.63% | 6.96% | **1.00e-05** | **1.12e-08** | **1.00e-05** | 52.92% | 59.49% |

