# OpenReview forum: "When LLMs get significantly worse: A statistical approach to detect model degradations"
_ICLR.cc/2026/Conference — ICLR 2026 Poster_

### Official Review · Reviewer_Fq3r · 2025-10-25

**Soundness:** 3
**Presentation:** 4
**Contribution:** 4
**Rating:** 8
**Confidence:** 1

**Summary:**

This paper investigates an important problem of testing whether Large Language Models (LLMs) degrade, e.g., after quantization or different serving environments. It proposes to leverage the principled McNemar's test to detect statistically significant performance degradation. On evaluation of open-source datasets, the proposed method is shown to be highly sensitive in detecting model degradation.

**Strengths:**

- Targets an important and practical problem of testing LLM degradation.
- Theoretically grounded approach using McNemar's test for degradation detection.
- Interesting empirical observations about LLM degradation with the proposed method.

**Weaknesses:**

- Evaluations can be expanded to more models other than LLaMA.
- Practical implications of the findings regarding regression testing could be further elaborated.

**Questions:**

This is an interesting paper targeting an important problem of testing LLM degradation. The theoretical analysis with McNemar's test is highly encouraging, as it provides a principled way to detect statistically significant degradation. This is a new and useful contribution to the field. However, I am not an expert in this field, especially regarding the theoretical aspects. I would be interested in the following questions:

- The evaluations are primarily on LLaMA models. How well does the proposed method generalize to other open-source LLMs?
- Will this method be applicable to small models as well, or is it specifically designed for large models?
- The "faster regression testing" part seems interesting. Could you elaborate more on the actionable advice on how to effectively use this method for degradation testing in practice?
- If we have access to model internals, can we use this method to pinpoint the specific components causing degradation?

---

> ### Author Response · Authors · 2025-11-18
>
> Dear Reviewer, when initially reading your review you gave a score of 8.
> We were surprised that you changed your review to a score of 4 now and it is
> unclear to us what the reason for that change was.
>
> We appreciate your feedback and believe that our revised version should address your concerns and questions
> and we hope that this gives you confidence to support acceptance of our work.
>
> Please consider our general reply above and also the revised version of the paper.
> We address your questions one-by-one below.
>
> > Limited importance of the targeted problem
> > The observations have limited practical implications
>
> We respectfully disagree. Research on LLM Inference is a huge field,
> and of immense practical and academic relevance. It is crucial to identify
> compression methods that are practically lossless and distinguish those from methods
> with real accuracy loss. Dutta et al (2024) and Kurtic et al (2025) also discuss this,
> but we show that their rules (while still helpful) are not supported by statistics, whereas our approach is.
>
> We also believe that our observation that a 0.3% accuracy drop can indeed be statistically
> significant has immediate practical implications. This is a rather surprising observation
> that was not known beforehand -- as also noted by other reviewers.
>
>
> > Evaluations can be expanded to more models other than LLaMA.
> [...]
> > The evaluations are primarily on LLaMA models. How well does the proposed method generalize to other open-source LLMs?
>
> We now evaluated on Mistral, GPT-OSS and also more tasks/frameworks,
> please refer to our general reply for more details.
> In particular on Mistral we also find that the w4a16 checkpoint is statistically
> significantly degraded despite having a drop of less than 1%.
>
> > Practical implications of the findings regarding regression testing could be further elaborated.
>
> The implication of a flagged regression is that we should expect degraded performance
> also when using the model on production workloads. It could then be an informed the decision to accep such a regression and still use the model or one might choose to resort to a less aggressive optimization
> approach. Furthermore, our regression tests can be implemented in continuous integration
> pipelines for example of vLLM or sglang to detect inadvertent model degradations, for example,
> when optimizing kernels.
>
> > Why is the technical contribution of applying an existing method to a new problem significant?
>
> The technical contribution we make is to properly formalize the setting of testing
> accuracy degradation across multiple benchmarks and to identify a suitable combination
> and generalization of existing statistical tools to properly solve this problem. Please
> see also our general discussion part c/
>
> > Will this method be applicable to small models as well, or is it specifically designed for large models?
>
> Our methodology is not specific to Large models. We just use "Large" language models as this is the general term for text-generating pre-trained models.
> But it would equally apply to Small language models, or even more generally machine learning models.
>
> > The "faster regression testing" part seems interesting. Could you elaborate more on the actionable advice on how to effectively use this method for degradation testing in practice?
>
> In practice our insights imply that we could create a smaller test dataset, just comprised of the examples that exhibit flips. This would  still result in almost as powerful tests, but significantly reduce the cost and duration of running the benchmarks. This is particularly relevant when including our methods in a continuous integration pipeline of a serving package.
>
> > If we have access to model internals, can we use this method to pinpoint the specific components causing degradation?
>
> From the accuracy regression we cannot directly pinpoint what exactly is the root cause for the degradation. Though in our cases it is clear that is the quantization. However, our contingency tables allow to quickly identify the tasks that are most affected and one can then dive deeper by investigating the cases where the optimized model failed.

---

> > ### Comment · Reviewer_Fq3r · 2025-11-18
> > **Regarding the response to review questions**
> >
> > In terms of the technical contributions, although I do not understand the full theoretical details of how the leveraged statistical test is applied to the degradation detection, the argument that "no out-of-the-box statistical solution for such a scenario" seems to be strong. Also, the cited review about Dutta et al. supports this point even further.
> >
> > I am still not confused about the practical implications of this work. Could you elaborate more on the point that "0.3% of accuracy drop has immediate practical implications"? Does it mean that a 0.3% accuracy drop is discernible with the help of the proposed tool, or does it mean that such a drop can severely harm the user experience? From my understanding, a 0.3% accuracy drop may not be critical; developers may be willing to slightly sacrifice model accuracy for inference speed.
> >
> > I am satisfied with the answers to the rest of the questions. Even though this tool cannot pinpoint the exact location of accuracy degradation, such a weakness is acceptable. One paper does not need to solve all the problems; clarifying such a limitation could inspire future work in this direction.

---

> ### Comment · Reviewer_Uko7 · 2025-11-18
> **Concerns regarding this review (from fellow reviewer)**
>
> I would like to raise concerns about this review from Reviewer Fq3r.
>
> As authors also mentioned, reviewer Fq3r changed their score from 8 to 4 without providing any justification. I don't think this behavior is respectful to other researchers.
>
> The review is also extremely low–quality and appears to have been generated by an LLM. All strengths, weaknesses, and questions are generic. I would give this review a strong reject if given a chance to review this review.
>
> As a fellow reviewer for this paper, I suggest this review be discarded.

---

> > ### Comment · Reviewer_Fq3r · 2025-11-18
> > **Clarification on the downgrading of paper rating**
> >
> > I appreciate everyone's efforts in reviewing this paper. I fully understand that the goal of the review process is to provide constructive feedback to improve the quality of the paper. I would like to respond to the concerns from Reviewer Uko7.
> >
> > Downgrading the paper rating from "accept" to "marginally below acceptance threshold, but would not mind if paper is accepted" is not a random choice but the result of careful considerations. The prominent concerns are about the practical implications of the targeted problem and the technical novelty of the paper.
> >
> > Regarding the practical implications, I am concerned about the usefulness of the paper. If model degradation is large, developers can easily notice such degradation even in the absence of the proposed technique; if the model degradation is small, which, at least from my perspective, is the major use case of the proposed method, the degradation is natural and expected. For instance, it is understandable that quantization can decrease the accuracy of the model. Developers expect such degradation and are typically willing to sacrifice the model accuracy for runtime efficiency. Why would it be useful to inform developers of the model degradation in such a scenario?
> >
> > In terms of the technical contributions, I appreciate the benefits of incorporating existing statistical methods, which offer theoretical confidence to model developers. However, I am not convinced that the technical contributions are significant enough. The leveraged theoretical methods are not new, and this paper applies an existing method to a new problem. I do not believe that the application of such a method is significant enough.
> >
> > These considerations led to my decision that the paper is "marginally below threshold," while also noting I "would not mind if this paper is accepted."
> >
> > Regarding the concern that "raised questions are generic," I would like to point out that the questions raised in the review are targeted to specific aspects of this paper and intended to improve the paper's quality. For instance, the concern over the practical importance of the method aims to help the paper articulate the implications of the method from a practical perspective. Questions about the actionable advice are intended to help practitioners gain insights into real-world model degradation issues. These questions are the result of careful reading and reflection on this paper.
> >
> > Nonetheless, as I mentioned in my expertise rating in the review, I am not an expert in this area. My review may not holistically reflect the true quality of the paper. It is better to consider all reviews, especially those from experts, to make a decision on the outcome of this paper.
> >
> > I believe that the peer review process should be based on facts and mutual respect. I would be happy to discuss further to improve my review if it does not meet the expected standards. I am also willing to cooperate with all reviewers and area chairs to make informed and fair decisions on this work.
> >
> > Thank you for your understanding.

---

> > > ### Comment · Area_Chair_f95J · 2025-11-18
> > >
> > > Thank you for the fast response, I would like to emphasize that I was referring to the rankings discussion, and wanted to highlight that we are all peers here, aiming to get good science shared (even if, in this massive system and burden, it is easy to forget it). It is great that you all are active, and I hope the discussion makes for a better paper. Just let's do it on science, be respectful, give the benefit of the doubt and in general treat others the way we would want our peers to treat us.

---

> ### Comment · Area_Chair_f95J · 2025-11-18
>
> I'll look into it, thanks for raising (Fq3r you are welcome to pitch in). (Uko7 why does it appear LLM-generated?).
> I believe the discussion that took place is hidden from authors and other reviewers? I agree the reviewer's update should have been earlier before it is shown to the authors, but I believe it was just an update by making a more thorough\another pass on the paper and thinking about some of the discussion points. Anyway, let's discuss how to make the paper better and how can we help the authors, this is the point of peer review. At the end, if the secondary goal of the review is unclear (whether the paper should be accepted), we can discuss it.

---

> ### Author Response · Authors · 2025-11-18
> **reply to reviewer Fq3r**
>
> Thank you for actively and timely engaging in the discussion. We are happy that you found our answers satisfactory, and we will try to address your last concern regarding the practical implications.
>
> >I am still not confused about the practical implications of this work. Could you elaborate more on the point that "0.3% of accuracy drop has immediate practical implications"? Does it mean that a 0.3% accuracy drop is discernible with the help of the proposed tool, or does it mean that such a drop can severely harm the user experience? From my understanding, a 0.3% accuracy drop may not be critical; developers may be willing to slightly sacrifice model accuracy for inference speed.
>
> The point of our discussion, and generally statistical hypothesis testing, is to decide whether an observed (accuracy) deviation based on finitely many examples is evidence of a *true* difference between two models or whether it is within the expected noise due to finite sample effects. Such a decision is difficult because there cannot be hard-threshold rules like "everything above x% constitutes evidence for a degradation, and everything below it does not". Consider, for example, Table 5 in the Appendix and the Llama 3.1 `FP8` checkpoint. This has an almost 1% accuracy drop on the MATH task, but our tools enable us to conclude that this can be safely attributed to evaluation noise. On the other hand, the 70B `KV-FP8` checkpoint has a pooled accuracy drop of 0.3%, and our tools clearly flag this as statistically significant.
>
> For practitioners this does have implications. While they don't need to worry about the 8B `FP8` checkpoint's 1% drop on MATH task, they need to worry about the 70B `KV-FP8` model because they have statistical evidence that the model is actually worse. Note that without our tools, a practitioner might act exactly the wrong way around.
>
> Now, you are absolutely right that they might still want to use the KV-FP8 model despite its statistically significant 0.3% drop because the performance or cost benefits outweigh the downsides. But this is then a deliberate decision based on *pros* and *cons*. On the other hand, for statistically lossless models, there is no evidence of *cons*, and thus there the trade-off decision is much simpler.
>
> Please note that we already discussed this in Section 5: "[T]he usefulness of an optimized model is not
> solely dependent on its accuracy but also its expected performance gains. Thus, a checkpoint can
> still be useful despite having *some* accuracy loss."
>
>
> We are happy to engage in further discussion to close any potentially remaining concerns about our work.
>
> PS: the more obvious case where you certainly do not want to ship a statistically significant 0.3% accuracy drop is when you intend to implement a *lossless* optimization. Say you are working on an optimized attention kernel variant, and want to check the functional correctness of it. Then on benchmarks you cannot accept any significant drop, even if it was just 0.3%. If you find such a drop this means that you have a bug in your code or some unintended numerical errors that you need to fix before it is truly lossless and working as intended. Without our statistical tools it is impossible to do this reliably.

---

> > ### Comment · Reviewer_Fq3r · 2025-11-20
> >
> > Thank you for clarifying this. Differentiating true degradation and expected noise with theoretical analysis is a strong indication of the practical implications of this paper. I have raised the score for the "contribution" and the overall rating.

---

> > > ### Author Response · Authors · 2025-11-20
> > >
> > > Thank you for the updated assessment. We appreciate your constructive engagement throughout the discussion and are glad we could clarify your concerns.

---

### Official Review · Reviewer_9xbY · 2025-10-31

**Soundness:** 2
**Presentation:** 2
**Contribution:** 2
**Rating:** 4
**Confidence:** 4

**Summary:**

The paper introduces a statistical framework for detecting significant performance degradations in Large Language Models (LLMs) after optimization techniques such as quantization. The paper adapts a one-sided exact McNemar test and offers three ways to aggregate over multiple benchmarks (pooled, max-drop with Monte-Carlo, Fisher) to statistically determine whether observed accuracy drops are due to true degradation or random noise. The result shows that even small drops (≈0.3%) can be detected as statistically significant and evaluates this on Llama-3.x variants run via lm-eval and vLLM.

**Strengths:**

The paper tackles an important gap in LLM evaluation, quantitatively distinguishing real degradations from noise, which is critical for optimization research. The framework is theoretically justified using McNemar’s test, with mathematical backing and variance derivations. The integration into the popular LM Evaluation Harness makes it useful for both academic and industrial research. The experiments cover different Llama models and optimization types (INT4, FP8, KV-FP8). The paper also gives a useful rule-of-thumb for compressing evaluation sets by removing examples that never flip in pilot runs, reducing evaluation cost while preserving signal.

**Weaknesses:**

•	The technical tool (McNemar 1947) and Fisher’s method are classical. The paper’s contribution is primarily in adapting these to LLM regression testing, making them one-sided, and packaging aggregation strategies. This is still valuable, but conceptual novelty is limited.

•	Main empirical claims revolve around Llama-3.x variants and specific serving stacks (vLLM; KV-FP8 issues noted). It is unclear how robust the conclusions are across other model families, in-house evals, or non-binary metrics. Without replication on other models/toolchains, significance to the broader community is limited.

•	While the paper sketches a generalization for non-binary scores, the actual implementation and experiments remain binary. Given how common non-binary metrics are for LLMs, a small worked example (pairwise wins/ties/losses with thresholds) would increase practical utility.

**Questions:**

1.	The paper needs to replicate the main findings on at least one other family (e.g., Qwen, Mixtral, DeepSeek) and another serving stack, demonstrating that 0.3–1.0% effects are robust.
2.	Beyond adopting the exact one-sided McNemar/binomial test, what is the new statistical contribution? Could the paper include a non-inferiority (TOST) module or sequential design that adapts sample size for power? How robust is the one-sided McNemar test when model scores are not perfectly binary (e.g., tasks with multiple correct answers or graded scoring)?
3.	The paper may run a prospective study showing how flip-focused trimming affects type-I/II error and bias across content strata, and propose guardrails. This directly addresses the validity of Recommendation 1.
4.	For non-binary metrics, please add a summarization or QA experiment (e.g., ROUGE or F1) and explicitly show the reduction to pairwise wins/ties/losses (or thresholds). The paper may also report one-sided p-values, effect sizes, and calibration to validate the procedure and demonstrate its generalizability and broader applicability.
5.	Could the paper specify and discuss the exact decision rule for flagging degradation? Does the framework flag when any one-sided aggregated test rejects at α (e.g., 0.05), when (\hat{\delta}>2%) or (\hat{p}*{\updownarrow}\ge 5%) (per Dutta et al., 2024), or only when both statistical and practical criteria are met? In the 70B KV-FP8 and 70B w8a8 cases where (\hat{p}*{\updownarrow}<5%), the framework still detects a performance degradation.
6.	The manuscript contains several language issues that impede readability (e.g., apostrophe misuse “its/it’s,” misspellings such as “assess”/“further,” incorrect conjunction “correct or false,” article usage “a similar situation,” verb form “build,” spacing/typography around symbols like “X, Y,” a duplicated footnote/URL, and inconsistent dialect choices like “while/whilst”). Could the paper (a) perform a thorough language edit to correct these issues, (b) standardize on a single dialect, and (c) resubmit with tracked edits or a brief editorial checklist summarizing the changes?

---

> ### Author Response · Authors · 2025-11-18
> **Reply to reviewer 9xbY (1/2)**
>
> We thank you for your generally positive assessment about the motivation and relevance of our work and for your detailed and technical feedback. While we will reply to your questions and concerns here, we kindly also ask you to check the general reply above and our revised paper. We believe that those largely address your weakness concerns and would appreciate if you could support acceptance of our paper.
>
> > Main empirical claims revolve around Llama-3.x variants and specific serving stacks (vLLM; KV-FP8 issues noted). It is unclear how robust the conclusions are across other model families, in-house evals, or non-binary metrics. Without replication on other models/toolchains, significance to the broader community is limited.
>
> In the revised version, we generalized our methods to non-binary scores by introducing a permutation test. We further added experiments on other model families (Mistral and GPT-OSS),  another serving framework (transformers), another evaluation framework (GPT-OSS evals), and a task with a non-binary score (TruthfulQA, Appendix D.4). Please see our general reply and the revised paper for more details.
> While in-house evaluations are not publicly accessible, the methodology and code are general and can be applied directly in such settings.
>
> > While the paper sketches a generalization for non-binary scores, the actual implementation and experiments remain binary. Given how common non-binary metrics are for LLMs, a small worked example (pairwise wins/ties/losses with thresholds) would increase practical utility.
>
> As described above, we managed to generalize our methodology to non-binary scores. In Table 11 and 12 we also added new experiments that show that a proper treatment of non-binary scores is much better (aka leads to more significant results) than thresholding or using win-rates with the binary tools.
>
> > The paper needs to replicate the main findings on at least one other family (e.g., Qwen, Mixtral, DeepSeek) and another serving stack, demonstrating that 0.3–1.0% effects are robust.
>
> Please see our previous replies. In particular our new results in Mistral show that a 0.87% accuracy drop of the w4a16 checkpoint is highly significant for model degradation.
>
> > Beyond adopting the exact one-sided McNemar/binomial test, what is the new statistical contribution? Could the paper include a non-inferiority (TOST) module or sequential design that adapts sample size for power? How robust is the one-sided McNemar test when model scores are not perfectly binary (e.g., tasks with multiple correct answers or graded scoring)?
>
> Please see our general reply regarding the methodological contribution. For non-binary scores we added the permutation test and solely recommend to use that with non-binary scores now.
> Regarding the TOST or sequential design: these are interesting directions. While they fall outside the scope
> of the present work, they could indeed be incorporated within our framework, and we consider them promising for future extensions.
>
> > The paper may run a prospective study showing how flip-focused trimming affects type-I/II error and bias across content strata, and propose guardrails. This directly addresses the validity of Recommendation 1.
>
> Could you please elaborate on this a bit more? Do you have concerns that after
> trimming our tests might not control type-I error anymore because we have
> already used some information of this sample and model combination and do
> not correct for this? Note that we obtain the flip probabilities at
> finite temperature with the base model. At finite temperature we expect the
> model to deviate much more from its temperature 0 behavior than any lossless
> optimized variant at temperature 0. We thus expect that the bias that affects type-I
> error would be negligible. Unfortunately, it is currently not clear to us how to
> design an experiment with an actual LLM to test this, i.e., what setting would
> we iterate over to estimate the type-I error rate.
>
> Regarding the type-II error, we expect that it slightly increases.
> As we show in the paper, we also remove a small portion of examples that would
> actually flip in the real degradation experiment and thus we loose statistical power. This is the price we
> pay for largely reducing the sample size and the cost of the actual evaluations.
>
> > For non-binary metrics, please add a summarization or QA experiment (e.g., ROUGE or F1) and explicitly show the reduction to pairwise wins/ties/losses (or thresholds).
>
> We added TruthfulQA with a ROUGE score variant in Table 12 where we used our new permutation test and compare it with reductions to binary scores.
>
> > The paper may also report one-sided p-values, effect sizes, and calibration to validate the procedure and demonstrate its generalizability and broader applicability.
>
> We kindly refer to Section C.2 where we did synthetic evaluations to provide insights into the general behavior of our tests (already in the initial submission).

---

> > ### Author Response · Authors · 2025-11-18
> > **Reply to reviewer 9xbY (2/2)**
> >
> > > Could the paper specify and discuss the exact decision rule for flagging degradation?
> >
> > As we discussed in the paragraph "Combined Decision" of Section 3.3, we recommend to flag degradation when either of our three tests results in a $p$-value below 0.05. We recommend to not use the flip probability or absolute accuracy degradation at all, as we don't have means to characterize their behavior under the null hypothesis.
> >
> > > The manuscript contains several language issues that impede readability [...] resubmit with tracked edits or a brief editorial checklist summarizing the changes?
> >
> > Thank you for the careful proofreading, we revised the paper and uploaded the new version.
> > We could not find “a duplicated footnote/URL” though. Could you please help us locate this? We checked all the footnotes and do not understand what you mean.
> >
> > We made the following changes:
> > - “it’s” → “its” end of section 2.1
> > - “fuhrter” → “further” in section 4
> > - “correct of false” → “correct or incorrect” in section 2.1
> > - “for similiar situation” → “for a similar situation” in section 1.
> > - "build” → “develop” in section 1.
> > - “built” → “build” in section 3.2
> > - “Let X,Y be” → “Let X and Y be” in section 2.1
> > - “whilst” → “while” in abstract, section 3.2, section 4
> > - “$\mathcal{X}$,we can define” → “$\mathcal{X}$, we can define” added a space in section 2

---

> > > ### Author Response · Authors · 2025-11-24
> > >
> > > Dear Reviwer 9xbY,
> > > just a gentle follow-up: if you have had a chance to look at our responses and the updated manuscript, we would very much appreciate any feedback you might have. We are happy to clarify any remaining questions. Thanks again for your constructive comments and for helping us improve the work.

---

> ### Comment · Reviewer_9xbY · 2025-11-27
>
> Thank you for the revisions. I appreciate the effort the authors put into addressing the earlier comments, especially the new experiments on additional model families and serving stacks, the extension to non-binary scores using a permutation test, and the clearer explanation of the decision rule. These updates strengthen the paper and make the overall contribution much clearer. I am raising the score for soundness, contribution, and the overall evaluation.
>
> p.s.: I wasn't able to find a place to edit the scores, but I would increase the soundness and contribution scores to 3, and the overall score to 6.

---

### Official Review · Reviewer_NK8t · 2025-11-01

**Soundness:** 3
**Presentation:** 3
**Contribution:** 3
**Rating:** 4
**Confidence:** 3

**Summary:**

This paper proposes a statistical framework to decide whether an optimized LLM has truly degraded relative too a baseline, rather than exhibiting harmless eval noise. The core is an exact one-sided McNemar test over per-item success/failure pairs fro the baseline vs optimized. This controls type-1 error and increase power by focusing only on disagreements (flips) between models. They derive theoretical analysis of test power, propose three aggregation methods for multi-task eval (pooled test, max drop test, Fisher's method), and provide implementation for the LM Evaluation Harness. Experiments on Llama-3.1 8B and Llama-3.3 70B with various quantization schemes demonstrate method can detect accuracy degradations as small as 0.3% while correctly not flagging theoretically lossless optimizations.

**Strengths:**

- well motivated as it addresses a real and understudied problem of rigorous statistical testing for LLM optimization
- first to apply McNemar's test to LLM degradation detection
- I like the practical contribution for LM eval harness which enables immediate adoption
- very interesting that even a 0.3% degradation can be detected with statistical confidence

**Weaknesses:**

- limited novelty: this is more about an application and implementation of statistical tests than methodological innovation
- I think some of the assumptions may not hold for many of the benchmarks today. they make an i.i.d assumption but real benchmarks (like MMLU) often stratify by difficulty, topic, or other factors which may violate independence. Additionally, the binary metric assumption is limiting as plenty of important metrics are continuous.
- I believe there is a limited scope of evaluation. the authors only evaluate on llama models (3.1, 3.3). no other model families are tested. they also only test quantization optimizations, but pruning, distillation, etc is missing.

**Questions:**

-  Given the three aggregation methods with different trade-offs, how should a practitioner choose?
- other than McNemar's test to answer if optimized model is statistically significantly worse than baseline, could you also use bootstrap confidence intervals for accuracy differences? or maybe permutation tests or bayesian model comparison (Bayes factors)? How do these compare in terms of power?

---

> ### Author Response · Authors · 2025-11-18
>
> We thank you for your initial assessment. We are glad to see that you consider our work a practically relevant new contribution. We would kindly ask you to read the general reply above and we clarify further points here. We hope that our revision and new experiments address your concerns, and would appreciate if you could support acceptance of our paper.
>
> > limited novelty: this is more about an application and implementation of statistical tests than methodological innovation
>
> Please see our general reply Section c/
>
> > I think some of the assumptions may not hold for many of the benchmarks today. they make an i.i.d assumption but real benchmarks (like MMLU) often stratify by difficulty, topic, or other factors which may violate independence. Additionally, the binary metric assumption is limiting as plenty of important metrics are continuous.
>
> Stratification in benchmarks (e.g., topics or difficulty levels) does not affect the validity of
> our paired tests, because the grouping is fixed and independent of the models’ predictions.
> Under the null, the only assumption required is per-item exchangeability of the paired outcomes,
> not global i.i.d., and heterogeneity across strata does not violate this.
> Additionally, our new permutation test (Appendix D) makes no distributional assumptions (beyond exchangeability under the null) and empirically confirms the same conclusions.
>
> Regarding the assumption about binary metrics, we managed to generalize to non-binary metrics via a permutation approach (see general reply and newly added Appendix D)
>
> > I believe there is a limited scope of evaluation. the authors only evaluate on llama models (3.1, 3.3). no other model families are tested. they also only test quantization optimizations, but pruning, distillation, etc is missing.
>
> Please see the general reply and the revised paper. In particular we included now also Mistral, GPT-OSS as well as pruned and distilled models in our evaluation. Please let us know if you have specific checkpoints in mind that we should further evaluate on.
>
> > Given the three aggregation methods with different trade-offs,
> how should a practitioner choose?
>
> In practice, we recommend flagging a degradation if either of our three p-values is below 0.05, see paragraph "Combined Decision" in Section 3.3 and Figure 3 in the Appendix.
> However, practitioners are of course free to choose more or less conservative thresholds when using our methods.
>
> > other than McNemar's test to answer if optimized model is statistically significantly worse than baseline, could you also use bootstrap confidence intervals for accuracy differences? or maybe permutation tests or bayesian model comparison (Bayes factors)? How do these compare in terms of power?
>
> Thank you for this great suggestion. We included the permutation test in the new revision, please also see our general reply.
> In particular, this approach a/ leads to almost exactly the same $p$-values on binary data and b/ directly allows for a generalization of our work to non-binary scores.

---

> > ### Comment · Reviewer_NK8t · 2025-11-18
> >
> > Thank you. this addresses my concerns, and I have raised my score

---

> > > ### Author Response · Authors · 2025-11-20
> > >
> > > Thank you for the updated assessment. We are glad that the revisions based on your feedback, as well as the discussion, helped address your concerns.

---

### Official Review · Reviewer_Uko7 · 2025-11-03

**Soundness:** 3
**Presentation:** 4
**Contribution:** 3
**Rating:** 6
**Confidence:** 4

**Summary:**

Authors argue that simply comparing aggregated accuracy scores of benchmarks are statistically flawed because it ignores the fact that two model evaluations are performed on the same set. Authors propose the testing framework based on the exact one-sided McNemar test. And further show that the testing framework can confidently flag model degradation as small as 0.3% to be statistically significant.

**Strengths:**

1. Authors challenge a fundamental approach in LLM evaluation benchmark where accuracies may not necessarily show the degradation of LLM capability. Currently the general practice is to consider difference within a certain threshold as statistically insignificant. Authors provide a more principled way to identify the flaws of the convention.

**Weaknesses:**

1. As a work studying variance/randomness, in the main experiment, it might be important for authors to report results across three runs.
2. Similarly, authors may want to reproduce an empirically lossless variant to rule out the hardware level randomness. (for example, use hf transformers backend and throw in the deterministic flag) Since we don’t necessarily know how the hardware-level randomness interact with the model-level randomness.

**Questions:**

1. Different inference engine backend (hf transformers, vllm, sglang) sometimes show different results on benchmarks. It would be helpful to test on different engines and tell if different engines may yield different results.
2. The method may be useful for measuring forgetting of language models after finetuning/continual learning. I encourage authors to explore other scenarios where degradation is an important topic (pruning, finetuning, etc.)

---

> ### Author Response · Authors · 2025-11-18
>
> Thank you for your positive initial assessment.
> We would kindly ask you to read our general rebuttal summarizing our main
> changes in the revised version.
> Here we reply to your remaining questions.
> We hope that our revisions address your remaining concerns.
>
> > As a work studying variance/randomness, in the main experiment, it might be important for authors to report results across three runs.
>
> In the initial submission we included a rerun of the exact same model for Llama-3.1 8B Instruct. There we found that the accuracy fluctuations through reruns are minimal, see `Baseline (rerun)` in Table 5 and Table 6 of the initial submission. Furthermore, the pooled test's and the fisher test's $p$-values are deterministic. For the max drop test, the noise is controlled by the number of simulation steps. We set this to 10^5, which leads to an estimation error of 10^4 for a $p$-value of 0.05. Thus, we conclude overall that the noise resulting from running only once is minimal. We would appreciate to not invest more compute to rerun further experiments.
>
> > Authors may want to reproduce an empirically lossless variant to rule out the hardware level randomness. (for example, use hf transformers backend and throw in the deterministic flag) Since we don’t necessarily know how the hardware-level randomness interact with the model-level randomness.
> > Different inference engine backend (hf transformers, vllm, sglang) sometimes show different results on benchmarks. It would be helpful to test on different engines and tell if different engines may yield different results.
>
> Thank you for this suggestion. This is exactly one application of our methodology that we have in mind. Say a model is available in transformers and a user wants to add it to sglang or vLLM, then they could use our methodology to verify the correctness of their implementation which would help to formalize and facilitate integration testing. To illustrate this we added a run of Llama-3.1 8B Instruct on `transformers` and added the results in Table 2. We find that there is no statistically significant difference between the two frameworks.
>
> > The method may be useful for measuring forgetting of language models after finetuning/continual learning. I encourage authors to explore other scenarios where degradation is an important topic (pruning, finetuning, etc.)
>
> We added experiments with a 2:4 sparse and distilled variant of Llama-3.1 Base and a pruned variant of GPT-OSS to our paper. Both cases show clearly significant accuracy drops. Please refer to the general reply and the revised paper for more details. This is in line with the literature, where for pruning approaches the reported accuracy drops are usually much larger than for quantization approaches.

---

> > ### Author Response · Authors · 2025-11-24
> >
> > Dear Reviewer Uko7,
> > We wanted to kindly check whether you had a chance to look at our replies and the updated manuscript. If you have any remaining questions or concerns, we would be very happy to clarify them. Thank you again for your thoughtful review and engagement so far.

---

> > > ### Comment · Reviewer_Uko7 · 2025-11-27
> > >
> > > Thanks authors for their rebuttal. I will maintain my original positive score.

---

### Author Response · Authors · 2025-11-18
**Generalized methods, new experiments and general reply**

We thank all the reviewers for their feedback and are happy to hear that the reviewers
agree that a proper statistical treatment of model degradations is an important
and understudied problem in LLM inference optimization. We appreciate the reviewers'
criticism and suggestions and have now addressed them in a revised version.

The main concerns about our initial submission shared among the reviewers are about
a/ our work focusing on binary scoring metrics
b/ limited scope of evaluation (only evaluating on quantized Llama 3.x models)
c/ limited novelty from a statistical perspective (building on the existing McNemar test)

We addressed points a/ and b/ through further theory and experiments
 and uploaded a revised version with changes highlighted (we will prepare a final and clean version after the paper decision, of course). We briefly discuss
 the new results as well as a general reply to concern c/, and clarify the remaining
 questions of each reviewer in separate replies.

We hope that this addresses the remaining concerns and are happy to clarify further questions.

## a/ Beyond binary scores (new Appendix 4)
We added **Appendix D**, where we introduce equivalents of our initial binary tests, where $p$-values are obtained via
permutations/bootstrapping. We show that on **binary data those permutation $p$-values are almost exactly the same as the ones
obtained with our initial binary tests**. Furthermore, the permutation tests natively work with non-binary metrics, like Rouge Scores or F1
 Scores. We added synthetic and real experiments (TruthfulQA) that show that our permutation tests should be used in such cases, rather
 than thresholding scores to binary, or using win-rates in the contingency table (as we recommended for simplicity in our initial submission).
 To reflect this, we also updated the discussion in Section 5, which is now a discussion of the generalization to non-binary scores,
 rather than a discussion of its limitation.

We will of course also release the code for that generalized test.

## b/ Limited scope of empirical evaluations (Section 4 & Appendix D.4)
The reviewers rightly criticized that our evaluations were focused on Llama 3.x models and quantization as optimizations. Furthermore, that we were only testing with vLLM. We have now added experiments on a **Mistral**-Small model (Section 4), which confirmed that less than 1% accuracy drop can be statistically significant (see the w4a16 results added to Table 2). Furthermore, we tested a **pruned and distilled** variant of the Llama-3.1 8B (base) model, also showing a statistically significant accuracy drop. In terms of frameworks, we included an experiment serving the model with the **transformers** library, showing that changing the framework is lossless in this case, as  it should be.

While our main experiments focused on the LM Evaluation Harness, we now also use OpenAI's evaluation suite to investigate **GPT-OSS**. In this case we indeed obtain non-binary scores (because the same sample is evaluated multiple times) and thus directly use our new permutation-based tests, see Appendix D.4. There we also test a **pruned** model with fewer experts in the MoE layers, which however, results in very bad performance.

These experiments strengthen the conclusions and demonstrate that our framework applies robustly across multiple models and inference stacks. If reviewers have specific cases that they think would be crucial to add, we are happy to take another look at those.

## c/ Limited statistical novelty

The evaluation of LLMs has two major difficulties. First, a model is usually
evaluated on
multiple benchmark tasks and, second, different models are evaluated with
exactly the same examples.
We are not aware of any out-of-the-box statistical solution for such a scenario.
Furthermore, in the LLM community, apparently a proper statistical treatment is not known
(Dutta et al, 2024; Kurtic et al, 2025).
You might also want to consider the reviews of Dutta et al, where the programm chairs
explicitly ask for a better statistical treatment https://openreview.net/forum?id=QVG7j29Sta&noteId=F8xc48nuZ1

Our contribution is not proposing a new test, but adapting and integrating classical
statistical tools into the specific multi-benchmark, paired-evaluation setting of LLMs,
 for which, to our knowledge, no principled solution existed.

We think it is crucial to popularize such tools, not only to detect too heavily optimized models,
but also to include them in continuous integration pipelines in large open-source
projects like vLLM.

In our paper, we thoroughly credit the historical work of McNemar and Fisher and do not
claim to make any groundbreaking contribution in terms of statistical methodology.

---

### Author Response · Authors · 2025-11-29
**Discussion summary and final author remarks**

Dear AC and reviewers,

We want to thank all the reviewers for their constructive feedback and for actively engaging with our rebuttal and the updated manuscript. We are glad that we got the chance to strengthen our paper (most notably by expanding the empirical evaluation and generalizing our method to non-binary scores) and to clarify all questions from each reviewer. **All reviewers now consider the paper (marginally) above the acceptance threshold.**

Following the exchange with reviewer Fq3R, we uploaded a final revision with two small changes to further motivate the need for our work:
- In Section 5, we now explicitly discuss *integration-testing* in inference engines as an important use case for our work (as discussed with Fq3R).
- In the related work section, we now include the (just published) NeurIPS oral of Yuan et al., which provides a detailed study of the non-determinism in LLMs and the resulting accuracy fluctuations, which are particularly severe for reasoning models.
While Yuan et al. carefully analyze the phenomenon, they do not provide statistical tools to distinguish noise from real degradations. Our work directly addresses this gap.


Yuan et al., *Understanding and Mitigating Numerical Sources of Nondeterminism in LLM Inference*, NeurIPS 2025

---

### Meta-Review · Area_Chair_999H · 2026-01-01

**Summary:**

**Paper summary:** The paper is well motivated to address an important gap in LLM evaluation, quantitatively distinguishing real degradations from noise. It proposes a statistical framework for detecting performance degradation in LLMs after optimization. The core contribution is adapting a one-sided McNemar test to LLM regression testing. The paper further proposes three aggregation strategies for multi-task benchmarks and provides an implementation integrated into the LM Evaluation Harness. Experiments on LLaMA-3.x models demonstrate that degradations as small as ~0.3% can be detected with statistical confidence.

**Review summary:** Reviewers generally agree the problem is important, the approach is well motivated and theoretically sound, and the work has strong practical relevance. Concerns focus on its generalization to other models, non-binary outcome, and real-world benchmarks such as MMLU, as well as its novelty of only adapting classical tools (McNemar’s test, Fisher’s method). The authors well solved most of the concerns except for novelty. Overall, I will suggest accepting as poster paper.

**Reviewer Concerns:**

There are three major concerns raised by reviewers:
- (1) experiments are limited to Llama-3.x models and mostly quantization-related degradations. Reviewers consistently request validation on other model families, optimization types (e.g., pruning, distillation, fine-tuning);
- (2) the framework relies on i.i.d. assumption and experiments only on binary outcome, it is not clear whether it holds for non-binary benchmarks and metrics, such as summarization or QA (e.g., ROUGE or F1). Reviewers concern its generalizability and broader applicability.
- (3) reviewers arfue its novelty, noting that the statistical tools (McNemar’s test, Fisher’s method) are classical. The contribution lies primarily in adaptation, one-sided framing, aggregation, and engineering rather than methodological innovation.

Authors well addressed concerns (1) and (2) by adding more experiments based on Mistral/GPT-OSS, QA dataset TruthfulQA and so on. Concern about novelty remains partially explained in rebuttal by highlighting the major contribution is not proposing a new test. Two reviewers who rated 4 would like to increase scores.

**Reviewer Scores:**

Score: 6; Weak / Borderline Accept (Poster)

---

### Decision · Program_Chairs · 2026-01-26

Accept (Poster)